# High-resolution impact-based early warning system for riverine flooding

Husain Najafi [1] ✉, Pallav Kumar Shrestha [1,2], Oldrich Rakovec [1,3], Heiko Apel [4], Sergiy Vorogushyn [4], Rohini Kumar [1], Stephan Thober [1], Bruno Merz [2,4] & Luis Samaniego [1,2] ✉

Despite considerable advances in flood forecasting during recent decades, state-of-the-art, operational flood early warning systems (FEWS) need to be equipped with near-real-time inundation and impact forecasts and their associated uncertainties. High-resolution, impact-based flood forecasts provide insightful information for better-informed decisions and tailored emergency actions. Valuable information can now be provided to local authorities for risk-based decision-making by utilising high-resolution lead-time maps and potential impacts to buildings and infrastructures. Here, we demonstrate a comprehensive floodplain inundation hindcast of the 2021 European Summer Flood illustrating these possibilities for better disaster preparedness, offering a 17-hour lead time for informed and advisable actions.

Flooding affects more people worldwide than any other natural hazard does[1] and represents one of the four key climate change hazards[2]. Approximately 1.81 billion individuals, constituting 23% of the global population, are found to be directly exposed to 100-year floods[3]. Anthropogenic climate change, inadequate investments of governments and the private sector, and cognitive biases in human perception and decision-making are usually blamed for disastrous flood impacts[4,5]. Since the 1990s, the observed number of record-breaking rainfall events has deviated substantially from a stationary climate and this deviation has occurred at an increasing rate[6]. The rarest rainfall events are projected to experience the most substantial relative increase in magnitude under future climate change[7]. Extreme and even unprecedented rainfall events, and the associated flooding, are thus expected to occur much more often than in the past. As flood preparedness and defences are often overwhelmed by such extremes, forecasting and early warning systems are perceived as crucial tools to safeguard human life and reduce monetary losses[8].

For decades, science and state agencies have been developing hydro-meteorological monitoring and forecasting systems[9,10]. Recent improvements in model resolution, process representation, parameterisation, data assimilation, and computational efficiency have advanced numerical weather prediction (NWP) and hydrological

forecasting, and early warning systems have benefited from that alike[11]. Efforts to enhance the monitoring of atmospheric variables and hydrological fluxes and conditions have also contributed to achieving more accurate initial conditions within the forecasting chain. However, the general public[12] and the media speculate why these scientific advances do not translate into reductions in socio-economic and human costs once a catastrophic event occurs - even in developed countries with advanced flood early warning systems (FEWS) as demonstrated by the floods in Western Europe in July 2021.

The components of the forecasting chain for a technologically advanced FEWS are depicted in Fig. 1. First, observed meteorological data is needed for generating hydrological initial conditions. The next component is the NWP system. The skill of NWP models is constrained by several factors, including intrinsic atmospheric chaos, errors in the initial conditions, the spatio-temporal resolution of the model, limited knowledge of physical processes, model errors, and limited computational power. However, with the steady progress of forecasting technology and skill over the past 40 years[13], NWP systems now provide improved quantitative precipitation forecasts because of the increased resolution to the scale of convective-permitting schemes (1–4 km), incorporating several sources of uncertainties and better representation of physical processes[14]. A substantial challenge in NWP

[1]UFZ-Helmholtz Centre for Environmental Research, Leipzig, Germany. [2]University of Potsdam, Institute of Environmental Science and Geography, Am Neuen Palais 10, 14469 Potsdam, Germany. [3]Faculty of Environmental Sciences, Czech University of Life Sciences Prague, Praha-Suchdol 16500, Czech Republic. [4]GFZ German Research Centre for Geosciences, Section Hydrology, Potsdam, Germany. ✉e-mail: husain.najafi@ufz.de; luis.samaniego@ufz.de

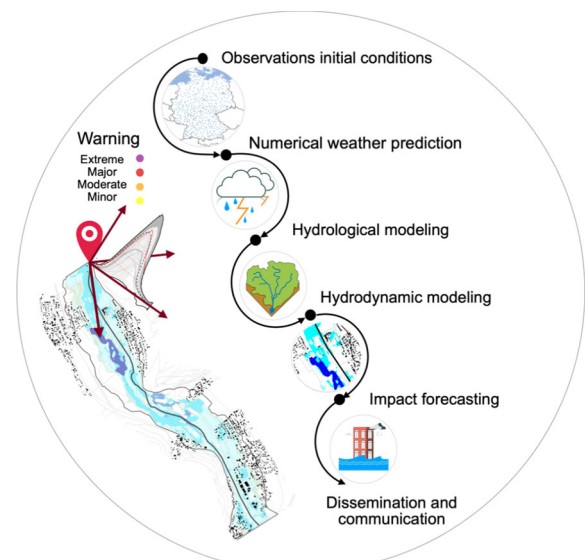

**Fig. 1 | A holistic end-to-end impact-based flood forecasting modelling chain.** The state-of-the-art flood early warning system is extended with components of quasi-real-time hydrodynamic and impact forecasting. Observational initial conditions are obtained based on data from ground, radar, satellite, and reanalysis. The Technology Readiness Level (TRL)[69] serves as a scale for evaluating the developmental stage and maturity of a technology. At TRL 1, the technology is in the initial scientific research phase, while TRL 9 signifies that the system has been successfully demonstrated in a real-world operational environment. Data sources: OSM rivers, roads and buildings: OpenStreetMap[60] contributors 2021, distributed under the Open Data Commons Open Database License (ODbL) v1.0. National German boundary: GADM. Meteorological stations (Deutscher Wetterdienst).

| Model/characteristic | Daily updates | Technology readiness level | Spatial Resolution | Ensemble member |
|---|---|---|---|---|
| Numerical weather prediction | 1 - 8 | 7 - 9 | 1 - 4 km | 10 - 51 |
| Hydrological forecasting | 1 - 8 | 7 - 9 | 100 m - 5 km | 10 - 51 |
| Hydrodynamic modeling and impact forecasting | Not yet operationalized | 4 - 6 | 1 - 10 m | - |

pertains to the uncertainties in precipitation forecasts, particularly for rare events[15]. These uncertainties propagate throughout the model chain and require quantification.

NWP model outputs are then passed to hydrological models to forecast discharge/water levels. Hydrological forecasting technology has also seen substantial progress. A decade ago, producing global hydrological forecasts from land surface models at a hyper-resolution of 0.1–1 km was viewed as a formidable challenge[16]. Achieving high-resolution hydrological forecasting is still ongoing within the research field. Delivering it would be possible with the availability of input data at high resolution and with the implementation of methods that derive seamless parameter fields as well as downscaled forcings and initial conditions[17]. Despite the widespread application of ensemble forecasting in NWP, ensemble flood forecasting is considered to be in its infancy even in countries with advanced operational FEWS[18,19]. This is mainly related to the challenges of transferring ensemble forecasts into operational decision-making and flood management[19]. Large-scale operational FEWSs that provide ensemble forecasts (e.g., the European Flood Awareness System-EFAS[20]) do not often satisfy the expectations of regional flood managers requiring hydro-meteorological forecasts at river gauge locations with high spatio-temporal resolutions and update frequencies[21].

Flood warnings are usually provided for river gauge locations. Extending flood forecasts from streamflow and water levels at selected river gauges to spatially distributed information on inundation, flow velocities and further impacts has been considered unfeasible for

many years[19] because of two main reasons: first, the extensive runtime required by fine-resolution (high-fidelity) hydrodynamic models to produce an ensemble forecast in real-time, and second, the lack of river cross-section data at a reasonably high resolution along the river network[22]. Despite the existing computational and operational challenges, flood managers need forecasted impact maps in real-time for issuing more targeted flood warnings and for better emergency responses[23]. By extending the forecast model chain with high-resolution (1-10 m grid size) hydrodynamic and impact forecasting, shown in Fig. 1, it would be possible to provide essential information downstream of the river gauge. For example, expected consequences of imminent flooding impacts, extending beyond traditional hazard data like river gauge water levels, affected assets and anticipated losses can be delivered. It holds considerable promises for enhancing disaster risk management by considering the physical characteristics of the event, as well as the affected socio-economic systems.

Local authorities and civil protection agencies benefit from impact forecasting, gaining actionable insights for initiating safety measures and evacuation protocols during floods. However, operational FEWS still need to integrate flood impact forecasting at the local scale of disaster management, particularly through the utilisation of 2D hydrodynamic modelling[24]. Table 1 provides an overview of the key components within existing state-of-the-art FEWS. Notably, both GloFAS[25] and EFAS employ an approach to inundation and impact forecasting, relying on the interpolation of pre-calculated flood hazard maps for a limited set of return periods[26]. This approach provides a rough estimate of potential inundation areas, and the so-produced flood maps are spatially inconsistent and do not retain continuity. This integration of the inundation prediction within operational FEWS presents two major challenges:

1. Computational Efficiency: Computationally efficient FEWS are imperative for promptly generating inundation and impact information, including associated uncertainties. Several studies have developed prototypes of flood forecasting modelling chains that include probabilistic flood inundation forecasting (see e.g. refs. 24,27,28). While high-fidelity models offer precision, they come with substantial computational demands. Strategies such as non-physics-based (simplified) methods[29] and model emulation, as demonstrated by Ivanov et al.[24] and Fraehr et al.[30], seek to strike a balance between computational efficiency and prediction accuracy. Sustaining prediction accuracy requires accounting for a wide range of flooding scenarios and inundation behaviours[30]. However, these approaches may encounter challenges when adapting to diverse flood scenarios or diverse landscape contexts[31]. Simplified methods, for instance, are particularly suitable for applications where dynamic effects play a minimal role, and the focus is primarily on the final or maximum flood extent and water levels[29]. Moreover, surrogate models may struggle when faced with inputs outside their training scope or complex, non-linear interactions among flood drivers[32]. Notably, they may also face difficulties accurately simulating unprecedented extremes compared to high-fidelity models[32].

2. Propagation and Representation of Uncertainties: Recent research demonstrates the potential usefulness of probabilistic forecasts for emergency managers facing real-world constraints. However, the exact impact of these forecasts on user decision-making remains unquantified[33]. The challenge resides in propagating the uncertainties along the entire forecast model chain and representing the uncertainty of impact indicators in a suitable way.

To address these challenges, the advancements in inundation and impact-based forecasting are demonstrated by comparing the common practice of pre-calculated flood hazard maps with our proposed forecasting chain. Utilising real-time forecasts as an extra layer enriches pre-calculated hazard maps by considering antecedent

**Table 1 | Existing state-of-the-art FEWS around the world**

| Platform /System | Scale | Atmospheric model | Atmospheric model resolution | Hydrologic models | Hydrologic model resolution | Forecast update | Ref. No. |
|---|---|---|---|---|---|---|---|
| GloFAS | Global | ECMWF-IFS | 18 km | HTESSEL-LISFLOOD | 0.1°–0.05° | 12-hourly | 25 |
| EFAS | Continental (Europe) | COSMO-LEPS/ICON/ICON-EU | 6.5–13 km | LISFOOD | 5 km | 6-hourly | 26 |
| Flood early warning system[a] | National (Germany) | ICON-D2/ICON-D2-EPS | 2.2 km | LARSIM, WAVOS | Spatial Units of 0.25–10 km² | 3-hourly | 21 |
| AHPS, HEFS, NWM[b] | National (USA) | AWIPS[c] | 3–25 km | CHPS | 100 m /250 m and 1 km[d] | 1–12 h | 18 |
| Hydrological Forecasting System (HyFS) | National (Australia) | ACCESS (Australian Community Climate and Earth-System Simulator) | 4–40 km | Unified River Basin Simulator (URBS), GR4J[e] | Semi-distributed | at least daily | 18,68 |

[a](Hochwasserfrühwarnsystem) The described model chains are implemented in flood forecasting centres in the German Federal States of Baden-Württemberg, Bavaria, Hesse, Northern Rhine-Westphalia, Rhineland-Palatinate, and Saarland[18]. Flood forecasting centres in other German Federal States are using similar approaches.
[b],[c]Flexible, specified by User (from HRR, NAM, GFS, RAM, and ECMWF).
[d]The National Water Model 3 provides 18-h deterministic short-range forecast for the contiguous United States (CONUS).
[e]SWIFT (GR4H-hourly) is used as part of the Short-term Water Information Forecasting Tools (SWIFT) hydrologic modelling package for 7-day streamflow forecast.

conditions[34]. In addition, fast hydrodynamic modelling captures real-time flood dynamics, overcoming the limitation of pre-calculated maps assuming a seamless connection between real-time forecasting models and static inundation and impact assessments, potentially leading to inaccuracies, especially for unusual flood events. Furthermore, these maps rely on several factors which might not be valid for all flood events[34]. We leverage fast and real-time hydrodynamic modelling while transparently communicating uncertainties for decision-makers. Our method, featuring dynamic simulation, provides crucial timing information for effective emergency responses. Additionally, it offers improved adaptability in flood hazard map resolution, particularly with high-resolution Digital Elevation Models (DEM), ensuring accuracy without sacrificing computational efficiency.

The need for impact-based warnings for disaster risk management has been addressed recently in various guidelines and studies[35,36]. For instance, the shift from weather forecasts and warnings to impact-based forecasts and warning services is outlined by the World Meteorological Organization (WMO) guidelines[37]. This shift also underlines the need for decision-making protocols tailored to align with the distinct dynamics of specific hazards, geographical locations, institutional capabilities, and cultural contexts. The initiative on impact-based early warnings is gaining global support, as more national hydro-meteorological services align their strategies and investments with this approach.

The effectiveness of an experimental impact-based flood early warning system is showcased in this study by utilising the catastrophic flood event that occurred in the Ahr River, Germany in 2021. During the July 2021 flood event, 134 people in the Ahr Valley lost their lives[21]. The total economic loss in Germany exceeded 40 billion EUR[38]. The return period of the event based on observed annual peak discharge gauge data between 1946 and 2019 and four historical floods between 1888 and 1920 is estimated to be about 8600 years[39]. The magnitude of the flood and its damage to buildings and infrastructure required the most extensive response and recovery operation in German history[38]. The German Weather Service (DWD) predicted a heavy precipitation event several days prior to the event[40]. In addition, the official hydrological forecasts indicated unprecedented water levels at several gauges. Post-event analysis has revealed that early warnings solely on hazard metrics such as maximum local rainfall depths or maximum water level at a gauge site resulted in misinformed actions, delayed responses, and at times, no action at all[38]. Local weather and civil protection officials underscored that their limited knowledge to understand the potential impacts of 150 mm or 200 mm of rainfall, or a gauge level of 6 m, prevented them from giving clear guidance on the specific problems or damage expected from the forecasted rainfall or water levels[38].

Here, we show how a state-of-the-art flood forecasting modelling chain can provide more sophisticated information, enhancing disaster preparedness. To illustrate the capabilities of our system, we provide a floodplain inundation hindcast ensemble for the 2021 European Summer Flood. The proposed approach allows for a more dynamic and responsive early warning system, offering enhanced insights into potential flood impacts. By employing high-resolution, object-based impact forecasting techniques, we are able to generate near-real-time flood inundation maps and other relevant impact indicators with associated uncertainties. This serves as a practical example to highlight the potential of our approach in accurately predicting and visualising flood impacts for better decision-making and preparedness in the face of such devastating events. It provides lessons that contribute to the improved management of future events and underscores why users need to put rare but severe events into perspective[33]. The current study serves as a proof-of-concept, laying the groundwork for the further development and testing of prototypes for such operational systems.

## Results

### Ensemble precipitation and probabilistic water level forecasts

Here, we use the DWD's latest NWP limited area ensemble prediction system (ICON_D2_EPS) for generating ensemble forecasts of water level for the event. The operational NWP ensemble prediction system generates 20 ensemble forecasts at a spatial resolution of 2.2 km. It considers different sources of forecast uncertainty arising from initial conditions and model error, in addition to the uncertainty in the boundary conditions for limited area ensembles[41]. For the hindcast experiment, ensemble forecasts were retrieved for every 3-h initialisation between 13 July 2021 (02:00 CEST) and 14 July 2021 (23:00 CEST), thus covering a window of opportunity of 47 h to 2 h prior to the flood peak.

Probabilistic forecasts are considered much more valuable than deterministic forecasts, especially for extreme and rare events[42]. Therefore, to evaluate the predictability of the flood event in Ahr Valley, with a catchment area of 746 km², the mesoscale hydrologic model (mHM)[43] is forced with 320 ensemble predictions (16 initialisations × 20 members) from ICON_D2_EPS to generate streamflow and water level predictions at the gauge Altenahr. The mHM has been evaluated as a prospective choice for continental-scale operational flood forecasting in Europe[44]. Ensemble medians and the water level predictions for all 16 initialisations are depicted in Fig. 2. Hydrological predictions for each initialisation is elaborated in Supplementary Fig. S1.

The Altenahr gauge was wrecked by the flood; therefore, water levels reconstructed by the responsible authority (Rhineland-Palatinate (RP) State Office for the Environment; LfU) are used for the evaluation of the ensemble water level predictions (refer to Fig. 2). The probabilities of exceeding warning levels are shown in Fig. 2 for each initialisation as well, assuming that all ensemble members have an equal likelihood[42]. The classification of official flood notification levels varies across Germany's federal states. In Rhineland-Palatinate, the categorisation of flood situations hinges on the concept of return periods. Specifically, flood occurrences with return periods equal to or exceeding that of a 50-year flood are labelled as extreme events. The 100-year flood (HQ100) serves as the critical benchmark for potential risks to life, property[42], and infrastructure.

Figure 2 displays a considerable variation in water level predictions among the ensemble members. This wide range of predictions can be attributed to the inherent uncertainties in the flood forecasting modelling chain, primarily stemming from ensemble precipitation predictions[42]. The precipitation forecasts derived from the ICON_D2_EPS (shown in Fig. 2) reveal substantial variations. These variations can reach up to 80 mm among distinct NWP ensemble members and across diverse forecast initialisations. The observed precipitation estimation for the event also exhibits uncertainty. The most realistic estimate indicates 119 mm of precipitation between the period 07/14 07:00 and 07/14 21:00 CEST[21]. This amount surpasses the ensemble median forecasts, and in some cases almost doubling them. Because this flood was an exceptionally rare event, and the calibration period has not had many such extreme events to tailor the model parameters,

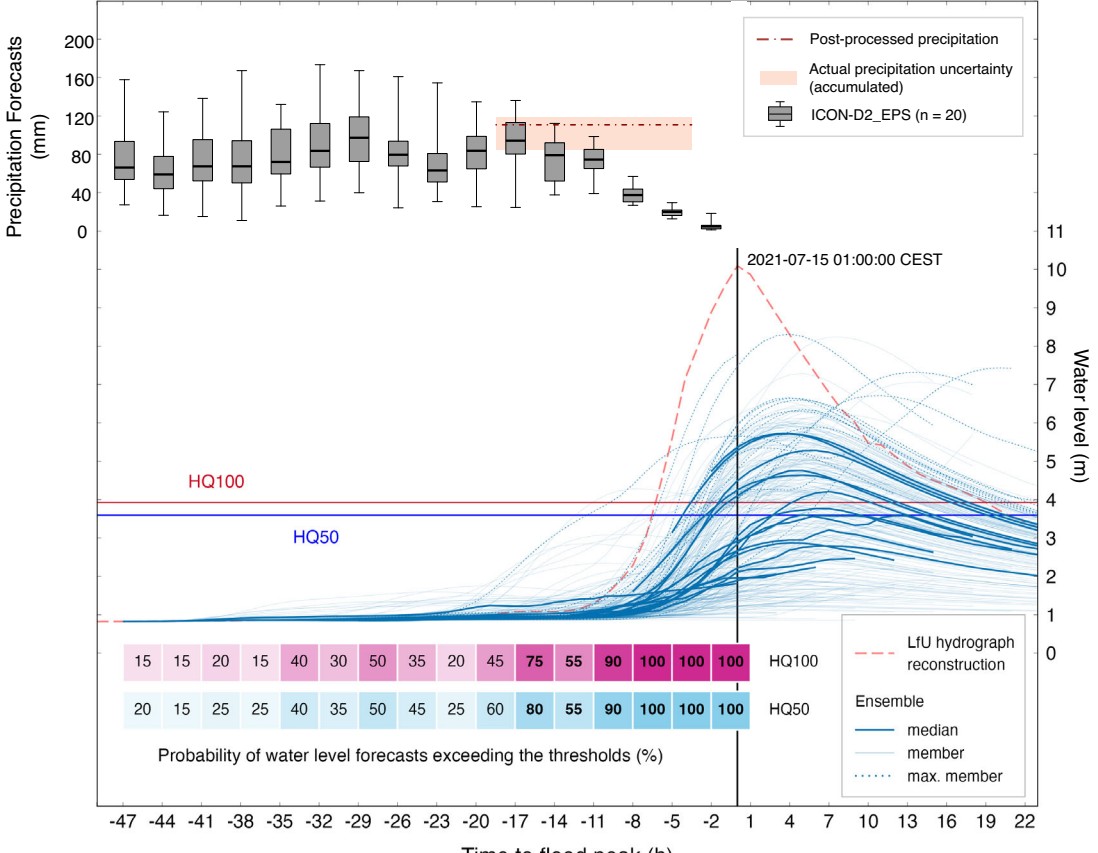

**Fig. 2 | Ensemble predictions of precipitation and water levels from the ICON_D2_EPS-mHM chain.** Ensemble predictions initialised every 3-h before the reconstructed flood peak at Altenahr gauge. The probabilities of exceeding the 50-year (HQ50) and 100-year (HQ100) flood thresholds are displayed for 16 forecast initialisations (See Supplementary Fig. S1 for more details). The range of 48-h areal precipitation forecasts for the Ahr basin is shown as whisker plots for each initialisation from ICON_D2_EPS. The whisker plots of precipitation forecast for each initialisation represent the minima, maxima, the bounds of the box (25 and 75 percentiles) and the centre (median) based on 20 ensemble members. The uncertainty of quantitative precipitation estimation for the event is shown for the target period of 07/14 07:00 to 07/14 21:00 CET[21].

precipitation amounts higher than 119 mm were necessary to accurately predict the flood peak. For these reasons, water level ensemble forecasts are substantially lower than the reconstructed water level at gauge Altenahr. Despite the discrepancies in the ensemble precipitation forecast, the primary focus remains on assessing the exceedance probability of the warning threshold as a key variable[45].

The expected precipitation amounts from high-resolution and convection-permitting NWP (ICON_D2_EPS) differentiate largely depending on the forecast time[21]. This uncertainty is propagated to water level predictions and finally to the probability of exceedance of warning thresholds. This complicates the task for flood managers, making it challenging to arrive at a confident decision[21]. For example, the probability of exceeding HQ100 increased by 30% from the forecast initialisation 20 h prior to the flood peak to that of 17 h but dropped by 20% in the next issued forecast.

For all water level forecasts issued within the lead time of 17 h to the flood peak, the probability of exceeding HQ100 is greater than 50% ($P_{WL>HQ100} \geq 50\%$) based on the ICON_D2_EPS-mHM forecast chain. Additionally, at the 11-h mark in advance (14 July, 14:00 CEST initialisation), the probability of a flood exceeding the HQ100 threshold surged to 90% (Fig. 2). This dramatic increase in probability further emphasises the urgency for appropriate flood response measures and is a confirmation of the adequacy of the modelling chain.

## Comparison between the official and experimental water level forecasts

Ten official deterministic water level forecasts were published by LfU within a time window ranging of 22 h–1 h prior to the reconstructed maximum level for Altenahr[21]. LfU forecasts ranged from 225 cm in the morning of the 14 July to 707 cm in the late evening. This wide range of predicted maximum levels illustrates the uncertainty associated with atmospheric forecasts and observation errors of the rain gauges and water levels at Altenahr gauge[21]. LfU uses the LARSIM water balance model[46] as an operational forecast model. The Ahr catchment is represented by 561 sub-basins in their model. The real-time forecasts on the 14 July 2021 were generated based on a LARSIM calibration from the period ranging between 1993 and 2016[21].

In the post-assessment report by LfU[21], an ensemble forecast was provided based on the ICON_D2_EPS for the 14 July 2021 (14:00 CEST) initialisation. The ensemble water level forecasts based on the ICON_D2_EPS – mHM are quite similar to the official forecasts for the same initialisation. Ensemble median water levels based on the ICON_D2_EPS – mHM were approximately 1 m lower than the deterministic water level forecasts of the LfU within the window of 5 h to the flood peak. Differences between water level forecasts may be due to the post-processing method used in the radar-adjusted quantitative precipitation estimate, the structural and parameter uncertainty, and the initialisation of the hydrologic model. In our proposed modelling chain, the LfU reconstructed hydrograph is used as a reference for the hindcast experiment.

## Lead-time maps and impact-based warning

Probabilistic water level forecasts at a gauge location do not provide sufficient information for emergency measures downstream. To address this shortcoming, the provision of lead-time maps to reach critical levels, along with high-resolution near-real-time inundation maps, and flow velocities are crucial and may ultimately save human lives and reduce socio-economic impacts[47–49].

Here, we demonstrate that near-real-time impact forecasting for floods is possible, even for comparatively small and fast-reacting rivers. The NWP-hydrologic forecasting chain is extended with the high-resolution (10 m grid) hydrodynamic model RIM2D, which proved to reliably simulate inundation for the Ahr valley[50]. In this study, the uncertainty along the modelling chain is considered, which is the added value compared to studies which have used only a single forecast (e.g., see Apel et al.[50]). The near-real-time forecasts of inundation

depth are compared first to HQ100 raster-based water depth map to identify regions with extreme flood hazard. For the grid cells, for which the water depth predictions exceeds HQ100, the lead time is calculated based on forecast outputs from hydrodynamic modelling. By running the ensemble inundation prediction, information on the most likely estimate of flood impacts can be derived from the ensemble mean. In addition, the ensemble members that have generated the minimum and maximum water levels can provide the uncertainty of inundation extent in each forecast initialisation.

In the presence of considerable uncertainties within the forecasting chain, effectively communicating forecast persistency is imperative for informed decision-making. Communication with local authorities should encompass the persistent impacts of flood forecasts, providing guidance for effective emergency response operations. Here, the selection of three consecutive forecast initialisations is considered. The lead time is calculated for each grid cell across consecutive forecast initialisations[11] when the water level surpasses the HQ100 threshold. To account for prediction uncertainty, we select ensemble members that produce the lowest and highest water levels at the gauge, in addition to the ensemble median, as well as the 25th and 75th percentiles. In Fig. 3a, b, lead-time maps for the ensemble median and maximum are presented for the river reach downstream of the Altenahr gauging station, covering multiple settlements. The high-resolution raster-based lead-time map shows a time window ranging from 6 h to 30 h, which could have been used for the most likely outcome (i.e., ensemble median). The maximum water level predictions indicate a lead-time map ranging from 24 h to 48 h before the forecasts exceed the HQ100 warning threshold. The predicted inundation extent from the ensemble median underestimates the actual flood extent mapped by LfU. The maximum ensemble member, i.e., one member out of 20, matches well with this estimate, (Fig. 3b). For this specific event, the assessment of the predicted inundation areas suggests that the flood extents generated by the maximum rainfall estimate from the ICON_D2_EPS model could closely resemble the actual conditions. This conclusion is confirmed by a recently published report[21], which shows that the observed precipitation was predominantly within the range of the maximum values of the ensemble forecast. Relying on lead-time estimation from a single forecast can extend the window of the opportunity for response, yet it may also elevate the occurrence of false alarms. Supplementary Figs. S2 and S3 present lead-time maps for median and maximum water levels without considering forecast persistence.

To validate the impact forecasts, we compared the affected buildings' footprint as well as road and railway lengths to those estimated by the Copernicus Emergency Management Service (CEMS) Rapid Mapping as a benchmark[51]. The service was activated by the German Joint Information and Situation Centre (GMLZ). The figures are compared for different ensemble members related to different percentiles (Table 2). For example, the maximum ensemble member, issued 47 h in advance of the flood peak, overestimated the inundated building footprint by 10% compared to Copernicus Rapid Mapping. Notably, the maximum ensemble member for several forecast initialisations closely aligns with the benchmark for several forecast initialisations. Our estimates for the number of affected buildings and infrastructure often turn out to be less severe compared to the post-event surveys. This aligns with the underestimation of water levels and consequently of the inundation areas.

## Enhancing flood forecast communication for informed decision-making and risk management

Decision-makers frequently encounter the task of issuing deterministic directives based on inherently probabilistic data[11]. Although FEWS capabilities may limit the provision of high-resolution flood impact forecasts and introduce uncertainties, effective communication of this information can still enhance user trust[52]. In the domain of emergency response, theoretical models and decision analysis methods abound,

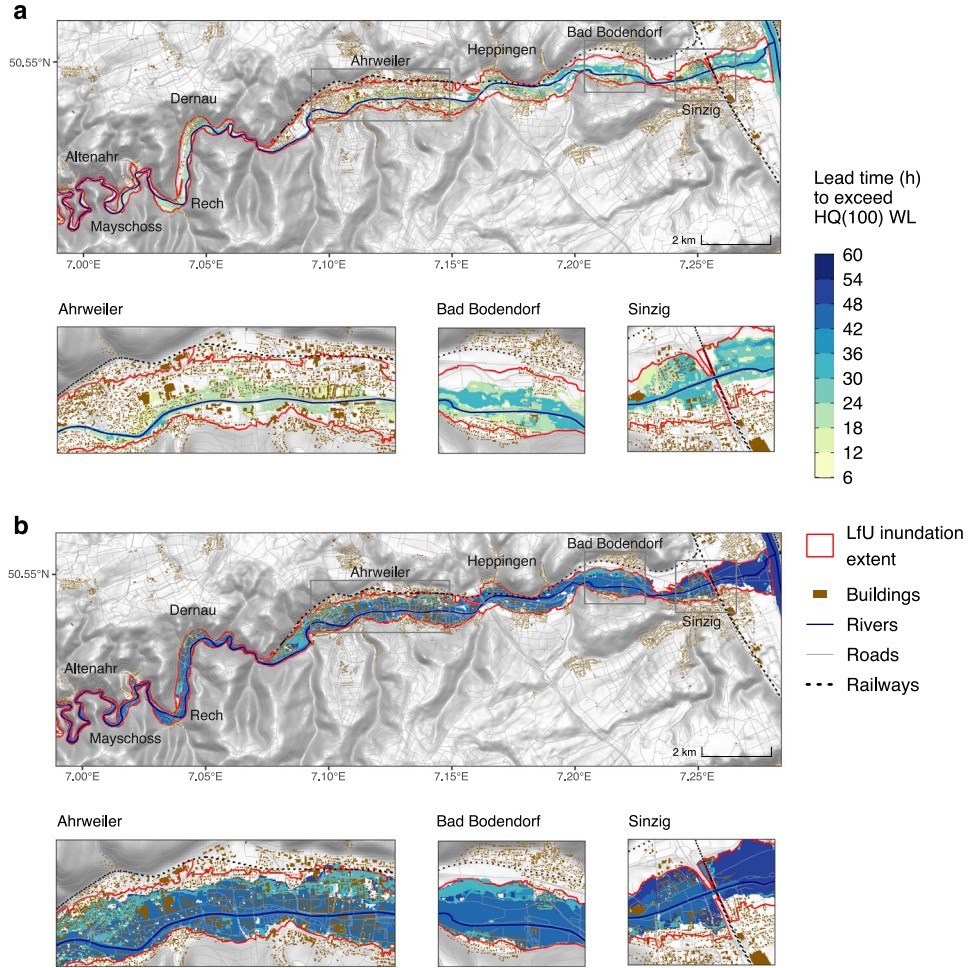

**Fig. 3 | The maximum flood lead-time warning based on the ICON_D2_EPS-mHM-RIM2D FEWS chain.** A maximum lead-time raster-based flood warning map is a geospatial representation that highlights the maximum available time for flood preparedness and response. The lead time is calculated downstream Altenahr gauge based on water levels (WL) exceeding HQ100. **a** The lead-time map derived from 16 ensemble median water levels (i.e., median over 20 members for each NWP initialisation). **b** The same but obtained with 16 maximum water levels. These lead-time maps are obtained when three consecutive initialisations exceed the HQ100 for a given 10 m grid cell. Please refer to the "Forecast persistency" section in the "Methods" section for additional details. The red extent delineates the inundation area mapped by the LfU of Rhineland-Palatinate. Supplementary data sources: OSM river, roads and buildings: OpenStreetMap[60] contributors 2021 distributed under the Open Data Commons Open Database License (ODbL) v1.0. Hillshade: DTM v0.3 (CC BY)[70].

with notable contributions like the Protective Action Decision Model (PADM)[53] and cumulative prospect theory[54]. In cases where official risk thresholds are not defined by relevant agencies, decision-makers often need to set their own probability thresholds that align with their specific needs and organisational goals, as illustrated by Fundel et al.[33]. In this respect, Fig. 4 provides a useful visualisation of how probabilistic information, based on lead-time, can support flood managers. The whisker plot visually represents the predicted inundated area downstream of the Ahr River, with an estimated coverage of 8.37 km². Visual representations like this effectively contextualise infrequent yet severe events, providing valuable perspective[33].

The convergence line in Fig. 4 falls short when compared to the 11.33 km² extent mapped by LfU due to the uncertainties inherent in forecasting rainfall, which subsequently impacts the predicted water levels and inundation extent. Nevertheless, the ensemble median inundation map has revealed that the affected area would potentially match or surpass the HQextreme level, which is the most extreme scenario of flood hazard mapping. Regarding Fig. 4, the ensemble median consistently surpassed the inundation areas of HQ100 and HQextreme by 20 and 17 lead hours, respectively. This time frame provides a potential warning lead time for preparation and response in the face of impending floods. For this particular event, it was

demonstrated that the maximum forecast ensemble member was more closely aligned with the post-event inundation area mapping compared to the median forecast. However, more events should be investigated to better understand how to use the full ensemble for decision-making. We advise flood managers to adjust the thresholds based on their daily experience in making warning decisions, as proposed by Fundel et al.[33], or in accordance with national regulations.

## Discussion

We demonstrate that recent advancements in hydrologic and hydrodynamic models and computational capabilities enable high-resolution flood inundation and impact forecasting within operational FEWS even for comparatively small and fast-reacting rivers. These forecasts encompass probabilistic inundation maps and identify buildings and transportation infrastructure at risk of flooding. Operational inundation and impact modelling provide much richer information on the space and time dynamics of flooding and its effects. Flood depth and flow velocities are not only available at a few gauge locations, but continuously and consistently in space. Time-varying characteristics such as lead time to specific depth thresholds or the rate of water rise can be provided during the course of the entire event.

**Table 2 | Comparison of impact-based forecasted damages to buildings, railways, and roads to benchmark**

| Infrastructure | Benchmark | Ensemble statistic | Forecast (Time to Flood Peak in hours) | | | | | | | | | | | | | | | |
|---|---|---|---|---|---|---|---|---|---|---|---|---|---|---|---|---|---|---|
| | | | 47 | 44 | 41 | 38 | 35 | 32 | 29 | 26 | 23 | 20 | 17 | 14 | 11 | 8 | 5 | 2 |
| | | | The ratio of forecast damage to benchmark (%) | | | | | | | | | | | | | | | |
| Building footprint | Copernicus Rapid Mapping | Max | 110 | 73 | 71 | 75 | 60 | 117 | 104 | 104 | 91 | 90 | 90 | 85 | 89 | 72 | 84 | 70 |
| | | 75p | 5 | 1 | 15 | 15 | 25 | 50 | 47 | 32 | 16 | 51 | 71 | 52 | 74 | 52 | 72 | 70 |
| | | Median | 1 | 1 | 1 | 1 | 1 | 4 | 12 | 14 | 1 | 15 | 44 | 29 | 57 | 41 | 70 | 70 |
| | | 25p | 1 | 1 | 1 | 1 | 1 | 1 | 1 | 2 | 1 | 3 | 20 | 1 | 46 | 32 | 69 | 70 |
| | | Min | 1 | 1 | 1 | 1 | 1 | 1 | 1 | 1 | 1 | 1 | 1 | 1 | 4 | 26 | 64 | 70 |
| | LfU | Max | 76 | 51 | 49 | 52 | 42 | 81 | 72 | 72 | 63 | 63 | 63 | 59 | 62 | 50 | 59 | 49 |
| | | 75p | 3 | 1 | 10 | 11 | 17 | 35 | 33 | 22 | 11 | 35 | 49 | 36 | 51 | 36 | 50 | 49 |
| | | Median | 1 | 1 | 1 | 1 | 1 | 3 | 8 | 9 | 1 | 10 | 31 | 20 | 40 | 29 | 49 | 49 |
| | | 25p | 1 | 1 | 1 | 1 | 1 | 1 | 1 | 2 | 1 | 2 | 14 | 1 | 32 | 22 | 48 | 49 |
| | | Min | 1 | 1 | 1 | 1 | 1 | 1 | 1 | 1 | 1 | 1 | 1 | 1 | 3 | 18 | 45 | 49 |
| Railways | LfU/Copernicus EMS Rapid Mapping | Max | 124 | 97 | 95 | 97 | 90 | 132 | 117 | 118 | 103 | 102 | 102 | 100 | 102 | 97 | 100 | 94 |
| | | 75p | 40 | 32 | 58 | 59 | 69 | 84 | 82 | 73 | 59 | 84 | 95 | 85 | 97 | 85 | 97 | 94 |
| | | Median | 29 | 29 | 29 | 29 | 32 | 39 | 55 | 56 | 30 | 57 | 80 | 70 | 90 | 78 | 94 | 94 |
| | | 25p | 29 | 29 | 29 | 29 | 29 | 29 | 29 | 36 | 29 | 38 | 65 | 29 | 82 | 72 | 94 | 94 |
| | | Min | 29 | 29 | 29 | 29 | 29 | 29 | 29 | 29 | 29 | 29 | 29 | 29 | 39 | 69 | 91 | 94 |
| Roads | LfU/Copernicus EMS Rapid Mapping | Max | 111 | 83 | 81 | 84 | 73 | 115 | 106 | 107 | 95 | 94 | 94 | 91 | 94 | 82 | 90 | 81 |
| | | 75p | 22 | 15 | 37 | 38 | 48 | 67 | 65 | 54 | 38 | 67 | 81 | 68 | 84 | 69 | 82 | 81 |
| | | Median | 14 | 14 | 14 | 14 | 15 | 22 | 35 | 36 | 15 | 37 | 63 | 51 | 71 | 60 | 81 | 81 |
| | | 25p | 14 | 14 | 14 | 14 | 14 | 14 | 14 | 18 | 14 | 20 | 43 | 14 | 65 | 54 | 80 | 81 |
| | | Min | 14 | 14 | 14 | 14 | 14 | 14 | 14 | 14 | 14 | 14 | 14 | 14 | 21 | 49 | 76 | 81 |

A 100% percentage indicates that the damage equals that of the benchmark while values exceeding (falling below) 100 signify an overestimation (underestimation) of the forecasted damage.

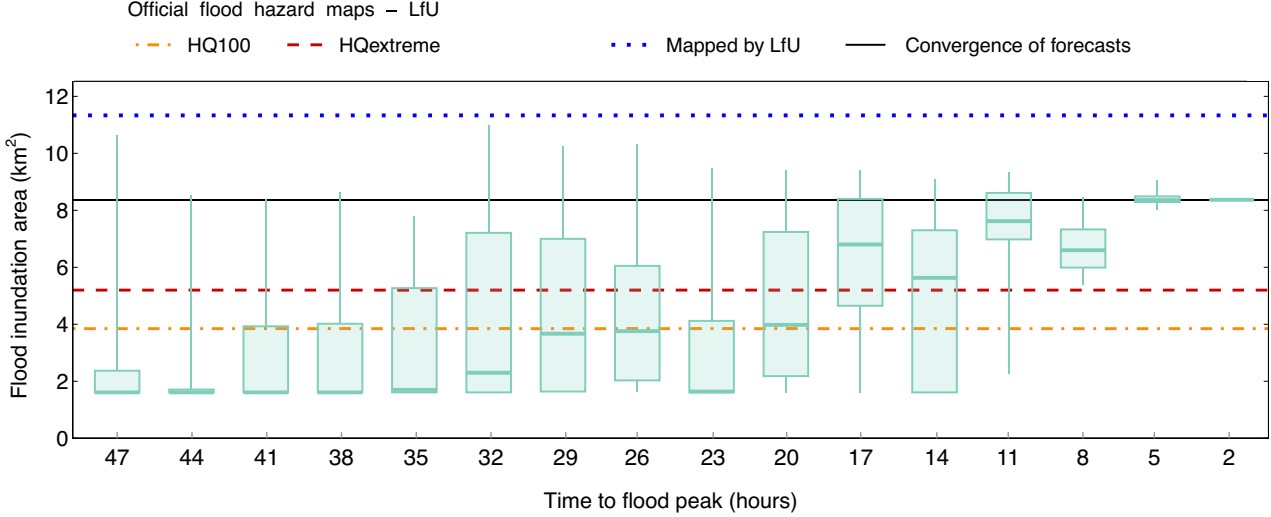

**Fig. 4 | Uncertainty representation of the forecasted inundated area downstream of the Altenahr gauge.** Uncertainty is quantified based on 16 initialisations issued 47 h to 2 h prior to the 2021 European Summer Flood. The uncertainty of the atmospheric forecast based on 20 ensemble members ($n = 20$) is propagated through the modelling chain to the hydrological and inundation prediction. The whisker plots of inundation prediction for each initialisation represent the minima, maxima, the bounds of the box (25 and 75 percentiles) and the centre (median) based on this ensemble. HQextreme represents the hazard map for the most extreme flood derived with a multiplication factor of a 100-year flood (HQ100).

At last, the prediction of affected buildings and critical infrastructure are compared against emergency mapping products derived from satellite data. Current satellite inundation maps, given their prioritisation of rapid mapping over quality, should not be regarded as absolute truth, leading to inherent uncertainties[55]. Using Synthetic Aperture Radar (SAR) for emergency mapping of floods may have some limitations, such as misclassification and timing issues. This highlights the need for caution and an acknowledgement of the upper limits of SAR-based flood detection methods when identifying affected areas and assessing damages[56].

The feasibility of the operational flood impact forecasting was demonstrated in the hindcast of the 2021 European Summer Flood event in the Ahr basin in this study. Several challenges, however, remain as we progress in adopting impact-based FEWS: (1) An increasing number of national hydro-meteorological services are investing in a paradigm shift from traditional FEWS to high-resolution, impact-based FEWS. However, implementing real-time services on a national scale poses challenges, given the trade-offs involving computational power, operational service scheduling, and data storage archiving. (2) Availability of quality data and computational resources is crucial for implementing near-real-time flood impact forecasting. Many regions, especially flood-prone areas, lack essential datasets such as high-resolution soil and terrain data, real-time meteorological observations, and high-resolution atmospheric forecasts. Continuous and long-term discharge measurements are also essential for flood monitoring, model calibration and warning threshold establishment. (3) NWPs still have uncertainties due to factors like ensemble size, model structure, and how small-scale convection processes are represented. Real-time precipitation can be underestimated and therefore needs to be post-processed for better accuracy. (4) The complexity of data integration and validation poses an additional challenge. The integration of workflow managers like ecFlow[57], complemented by a user-friendly graphical interface, streamlines the scheduling of operational services. This not only enhances user engagement and accessibility but also contributes to the optimisation of service delivery in real-time hydrodynamic modelling and forecasting. (5) Evaluating the performance of FEWS can be difficult, especially when hindcast data is not available and NWP models have limited operational history. Ensuring the reliability of FEWS is critical to respond quickly to predicted events. It helps avoid the cry wolf effect, where too many false alarms make people and authorities less likely to act promptly during real flood threats. (6) The prediction of rare, extreme flood events with return periods of more than a century is a challenge due to the limited data available, which emphasises the need for comprehensive training of flood managers. (7) The introduction of real-time impact-based warnings should go along with the development of specific customised warning messages, action instructions and emergency decisions. In order to tackle this problem effectively, interdisciplinary cooperation with social and psychological sciences is required. (8) In a world where the likelihood of unprecedented rainfall and subsequent flooding is increasing, impartiality in the communication of information is critical. Ongoing calibration of the hydrological component of operational FEWS is important to better anticipate flood events. Moreover, there is a growing demand to account for the most extreme events to avoid surprises of megafloods similar to the 2021 European Summer Flood[58]. The shift in thinking beyond national flood risk assessment and the removal of cognitive biases are necessary to prevent unexpected surprises[4,58].

Finally, more attention needs to be paid to the effective communication of forecast uncertainties. Uncertainties need to be propagated along the entire forecast chain delivering the plausible ranges of flood impact indicators. We believe that better-informed decisions can be made given transparently presented uncertainties rather than single deterministic values. Future studies are needed to find out how the proposed impact-based FEWS can be used for better communicating the flood impacts to users, decision-makers and the public. A subsequent investigation could involve seeking input from decision-makers regarding their preferences for ensemble ranges.

## Methods

### Extended warning chain

The extended warning chain (ICON_D2_EPS-mHM-RIM2D) is illustrated in Supplementary Fig. 4. This model chain produces high-resolution impact forecasts indicating inundation depth and flow velocity at buildings and infrastructure. The four components of this chain are described below.

**Meteorological inputs.** The regional ensemble prediction system ICON-D2 EPS provides operational forecasts for a 48-h forecast horizon, covering the entire German territory. High-resolution forecasts of ICON-D2 (2.2 km) are initialised every 3 h with a convection-permitting model set-up suitable for early warning of local heavy rainfall events. Hydrological initial conditions are derived from near-real-time radar-adjusted gridded hourly precipitation data provided by the DWD. The gridded fields of temperature were generated by using the External Drift Krigging (EDK) method[59] using variograms derived from DWD station observations.

**Ensemble hydrological forecasting.** Streamflow and water level forecasts are generated based on mHM at a resolution of 1.1 km. The mHM uses multiscale parameter regionalisation for estimating distributed parameter fields[43] and is forced with real-time forecasts from DWD-ICON_D2_EPS for hindcast evaluation and hydrological predictability of the Ahr flood.

**Hydrodynamic forecasting.** The hydrodynamic model RIM2D was setup and validated for the 2021 flood event in the Ahr valley[50]. Flood inundation depth for HQ100 was mapped first by running RIM2D. Then, the lead time of water level forecasts exceeding the HQ100 level was calculated for each raster cell at 10 m resolution downstream of the gauge Altenahr. The locations of buildings, roads and railways were extracted from the OpenStreetMap (OSM) layers. Hydrodynamic forecasts are triggered only upon reaching or exceeding pre-established warning thresholds customised for selected percentiles based on the user's specific interest. This automated trigger mechanism enhances the responsiveness and adaptability of the system accommodating real-time services easier. The RIM2D simulations are executed on the Graphical Processor Units (GPUs) to achieve high computational performance. Each ensemble run is allocated to a single GPU device allowing for parallel processing. While 20 ensemble members are available, our real-time forecasting focuses on selected percentiles with respect to peak discharge at the upstream boundary (minimum, 25%, median, 75%, and maximum). This approach ensures timely forecasts every 3 h and is able to accommodate larger ensembles if needed.

**Quantitative impact forecasting.** Several criteria can be provided for impact forecasting including object-based forecasting (e.g., building footprint), and the length of roads and railways. This information is calculated based on the synthesis of data extracted from open geographic databases such as OpenStreetMap[60], and hydrodynamic forecasting outputs.

**Copernicus EMS Mapping products.** The Copernicus Emergency Management Service (CEMS) employs satellite imagery and additional geospatial data to respond to natural disasters, including floods. CEMS offers a variety of products that provide insights into the impact and reach of the event, including overall flood extent and detailed assessments of damage severity[51]. It provides information on affected buildings and infrastructures based on several detection methods such as semi-automatic and automatic extractions. We utilised the standard spatial datasets (vector data) from CEMS, which are publicly available free of charge[51].

**Comparison between forecasts of inundated building footprint with a benchmark.** In this research, we analysed the number of affected building footprints, as well as the total lengths of roads and railways from RIM2D inundation forecasts by benchmarking them against established data sources. Our study utilised datasets from OSM and CEMS. The CEMS dataset provides valuable information on the extent and severity of flood impacts based on damage grades (ranging from damaged to potentially damaged and destroyed), and their spatial

distribution. The processing of this data involved several key steps: (1) CEMS data points corresponding to OSM building centroids were linked to the respective OSM building footprints; (2) in cases where multiple CEMS data points reported damage to the same OSM building, the OSM footprint was counted only once to eliminate duplication; and (3) CEMS data points lacking corresponding OSM building polygons were excluded from the analysis. We leveraged OSM data to furnish building footprints for structures affected according to the CEMS dataset. The processing and analysis were carried out using a combination of Python and R scripts, encompassing geospatial matching, damage statistics, and assessments of spatial distribution. Moreover, we compared the predicted inundation impact on building footprints for each initialisation with the total building footprint within the flood extent, as mapped by the LfU. This comparative analysis allowed us to thoroughly evaluate the accuracy and reliability of our predictive models.

### Hydrological model setup and calibration

The mHM setup used in this study is based on the BUEK200 soil dataset[61]. Soil layers are vertically discretized in four layers (0–5, 5–25, 25–60, and 60-cm · variable) in the mHM. More details regarding the mHM setup are described by Boeing et al.[62]. The corresponding mHM global parameters were calibrated using the Dynamically Dimensioned Search (DDS)[63] algorithm with 500 iterations, against observed hourly time series of river discharge at Altenahr gauge. A detailed description of the procedure for calibrating mHM can be found in Rakovec et al.[64]. In the present case, we considered a 10-year simulation period (1.1.2011–31.12.2020) with five years of warm-up; thus the July 2021 flood peak was excluded from the calibration exercise. Hourly RADOLAN grids of precipitation[65,66] are adjusted to 24-h total precipitation[67] and used for model calibration. The mHM historical performance is provided in Supplementary Fig. S5.

### Computational resources and data requirements

The implemented near-real-time flood impact forecasting chain is applicable to another region around the world contingent upon the availability of specific quality data and appropriate computational resources. To ensure effective operation, it is necessary to generate frequent NWPs of precipitation and temperature. Near-real-time access to hourly precipitation and temperature, observations is required for the regular reinitialisation of the mHM model. For the RIM2D hydrodynamic model, a high-resolution DEM and land-use information is required. We tackled the computational challenge inherent in real-time inundation forecasting through the utilisation of the massively parallelised Graphical Processing Units (GPUs)[50]. Using the state-of-the-art NVIDIA Tesla P100 device, we achieved a 22-min runtime for a 48-h event simulation (one ensemble member) for the entire domain of about 30 km river length with a spatial resolution of 10 m by 10 m. We would like to emphasise here that all the underlying datasets and modelling tools which have been used in this study are available freely. To develop a similar system in other regions, high-resolution terrain information (DEM) along with morphological datasets (e.g., soil, vegetation, etc) would be needed. Additionally, access to near-real-time meteorological forcings and river gauge station data for model calibration can be acquired from responsible agencies. To this end, the growing availability of remote-sensing and satellite-based information can provide additional opportunities to reliably establish the FEWS in data-scarce regions.

### Definition of forecast persistency

The probability of exceedance of a predefined warning threshold can rapidly change with subsequent forecast initialisations. A definition is provided for the forecast persistency in the forecast information across different initialisations. Once three consecutive forecast initialisations show water levels above HQ100 for a given grid cell, the time span between the time point of the forecast initialisation and model time step corresponding to $WL \geq HQ100$ for the third forecast is calculated as the lead time. The definition of lead time with and without confidence is provided in Supplementary Fig. S6. Selecting the ideal number of forecast initialisations to establish forecast persistency can be determined by balancing the frequency of operational NWPs and the required preparedness time.

### Reporting summary

Further information on research design is available in the Nature Portfolio Reporting Summary linked to this article.

## Data availability

REGNIE, RADOLAN data and location of meteorological stations, all from DWD, are freely available for research at the Open Data Portal (https://opendata.dwd.de, last access: 9 May 2022). DWD weather forecasts (ICON-D2-EPS) are available at Pamore (PArallelMOdel data REtrieve from Oracle databases) after registration (https://www.dwd.de/EN/ourservices/pamore/pamore.html, last access: 29 Nov 2022). The Digital Elevation Model with a 10-metre grid, is provided by ©GeoBasis-DE/BKG 2022 and is available under restricted access for free use. Access can be obtained by (https://gdz.bkg.bund.de/index.php/default/digitale-geodaten/digitale-gelandemodelle/digitales-gelandemodell-gitterweite-10-m-dGm10.html), Bundesamtfuür Kartographie und Geodäsie, 2022. OSM river network and buildings are available from ©OpenStreetMap contributors 2021, distributed under the Open Data Commons Open Database License (ODbL) v1.0. The OpenStreetMap data are open source. DTM v0.3 (to derive hillshades) can be obtained at https://opengeohub.org/datasets/european-digital-terrain-models-eu-dtm/. The hourly data from the gauges in the Ahr basin were kindly processed and provided by Michael Göller, Landesamt für Umwelt, Rheinland-Pfalz (https://wasserportal.rlp-umwelt.de/servlet/is/8181/). Streamflow gauge data and mHM output are provided for calibration of the hydrological model. Flood hazard maps are based on (https://geoportal.bafg.de/karten/HWRM_Aktuell/#). We used the flood hazard maps from Rhineland-Palatinate (https://hochwassermanagement.rlp-umwelt.de/servlet/is/200041/) accessed on 23.06.2022. This flood hazard map could have been updated after the 2021 flood event. The mapped inundation extent was provided by the Landesamt für Umwelt (LfU) Rheinland-Pfalz (Dr.Thomas Bettmann). The dataset for generating flood impact forecasting can be accessed under this link https://www.ufz.de/record/dmp/archive/14607/en/. The supplementary data for Figs. 2, 4 and S5 are provided with this paper. The shapefiles from the German national boundary and its neighbouring countries can be accessed under this link: https://gadm.org/maps/DEU.html. Source data are provided with this paper.

## Code availability

The mHM v5.11 and EDK codes can be found at https://git.ufz.de/mhm/mhm and https://git.ufz.de/chs/progs/edk_nc. RIM2D is available for non-commercial use at https://git.gfz-potsdam.de/hydro/rfm/rim2d. The codes and algorithms to generate figures provided in this study can be accessed under this link https://www.ufz.de/record/dmp/archive/14607/de/.

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

## Acknowledgements

This work is partly funded by Helmholtz-Climate-Initiative (HI-CAM) in the Helmholtz Associations Initiative and Networking Fund and by the BMBF (German Federal Ministry of Education and Research) project KAHR (01LR2102F). Open access funding enabled and organised by the project DEAL. We kindly acknowledge the German Weather Service (DWD), the European Environmental Agency (EEA), the Federal Institute for Geosciences and Natural Resources (BGR), the Federal Agency for Cartography and Geodesy (BKG), the European Space Agency (ESA), the Global Runoff Data Centre (GRDC), OpenStreetMap as data providers. The scientific results have been computed at the High-Performance Computing (HPC) Cluster EVE, a joint effort of both the Helmholtz Centre for Environmental Research - UFZ and the German Centre for Integrative Biodiversity Research (iDiv) Halle-Jena-Leipzig (http://www.idiv-biodiversity.de/). We would like to thank the administration and support staff of EVE who keep the system running and support scientific computing needs: Guido Schramm, Toni Harzendorf, Tom Strempel and Lisa Schurack from the UFZ, and Christian Krause from iDiv.

## Author contributions

H.N., L.S. designed this study and the forecasting system. H.N. coded the forecasting system with inputs from P.K.S., H.A. P.K.S. and S.T. added functionality in the mHM code base for sub-daily analysis. H.N. wrote the initial draft and conducted the hydrological model runs. O.R. established the mHM setup and performed parameter estimations. H.A. performed the hydrodynamic simulations with RIM2D. H.N. and P.K.S. conducted an analysis and produced graphs. H.N., P.K.S., O.R., H.A., S.V., R.K., S.T., B.M., and L.S., contributed to interpreting results and the revision of the manuscript.

## Funding

## Competing interests

The authors declare no competing interests.
