## [Peer Review File · Nature Communications]

High-Resolution Impact-based Early Warning System for Riverine FloodingEditorial Note: Parts of this Peer Review File have been redacted as indicated to remove third-party material where no permission to publish could be obtained.

REVIEWER COMMENTS

Reviewer #1 (Remarks to the Author):

The paper titled "On the importance of high-resolution and real-time impact forecasting for flood early warning" by Najafi et al tackles a very important topic. However, I have major concerns about the focus and novelty of the paper.

Comment 1

The paper states in the abstract: "Here, we demonstrate that better informed crisis management and significant loss reduction can be achieved with a FEWS that includes extended components for inundation modeling and impact forecasting." However, the paper contains no such demonstration. The paper presents a modelling chain for ensemble forecasting of flood inundation and impacts. It does not present a framework on how the ensemble information can be used in decision-making during crisis management, achieving loss reduction, for example, how many lives might have been saved in the case study.

In the case study, the maximum forecast ensemble member was shown to better align with the final observation than the median forecast. In general, however, how should the decision-makers use the full ensemble distribution? In another event, the maximum forecast member many turn out to be much too high compared with observation. Therefore, using the highest forecast member to guide decision-making may not be a good approach.

On a minor point, equating the largest member from an ensemble of size 20 to 5% chance of exceedance can be misleading as the largest member will be subject to sampling error. One would need a much larger ensemble size to estimate the 95% quantile.

Comment 2

A prototype of flood forecasting modelling chain that includes ensemble/probabilistic flood inundation forecasting is not new. For example,

- Gomez et al (2019) Skill of ensemble flood inundation forecasts at short- to medium-range timescales, *Journal Hydrology*, <https://doi.org/10.1016/j.jhydrol.2018.10.063>
- Schumann et al (2013) A first large-scale flood inundation forecasting model, *Water Resources Research*, 0043-1397/13/10.1002/wrcr.20521
- Ivanov et al (2021) Breaking Down the Computational Barriers to Real-Time Urban Flood Forecasting, *Geophysical Research Letters*, <https://doi.org/10.1029/2021GL093585>

I do, however, acknowledge that ensemble flood impact forecasting is relatively new.

I question several critical statements made in the paper:

"The near-real-time impact forecasting study performed here is a proof-of-concept that can be implemented in any other region around the world with reasonable run times." What is the basis for this generalisation? It is widely known that the large computational demand for running high-resolution hydrodynamic models poses a real barrier for producing ensemble flood inundation forecasts. The authors have not explained how they solved the problem by using the hydrodynamic model RIM2D. Is RIM2D much faster than other hydrodynamic models? What is the computational requirement of RIM2D for a range of catchments/floodplains?

"Extending flood forecasts from streamflow and water levels at single river gauges to spatially distributed information on inundation, flow velocities and further impacts has been considered unfeasible for many years because of the extensive runtime of hydrodynamic models for an ensemble forecast in real-time and the lack of river cross-section data at a reasonably high resolution." As my comment in the last paragraph, the authors have not explained how they overcome the problem.

"The RIM2D simulation, with its short run-time of approximately 22 minutes for a 48-hour model simulation, ...". The authors state that 20 ensemble members are available every 3 hours. With a run time of 22 min, they cannot be run after each other on a single computer, as this would take longer than 3 hr. The authors have not described how they will set up a system that can run all 20 ensembles every 3 hours in real-time. What computers and how many are needed?

"While running a full ensemble-based hydrodynamic modeling runs in near real-time can still be limited with respect to ensemble size and catchment areas, it can provide information on uncertainty and probability of streamflow/water level exceeding critical thresholds for early warning". The logic in this sentence is unclear to me.

I should mention that there is active research in the development of fast emulators of high-resolution hydrodynamic models, for the purpose of ensemble flood inundation and impact forecasting.

Comment 3 (minor)

In general, the writing needs much improvement, especially on the balance of content. There is not enough material/emphasis on the "new" work, that is, ensemble flood inundation and impact forecasting (and the use in decision-making). The conclusions have a mix of general conclusions and conclusions specific to the case study; it is not always clearly described.

Reviewer #2 (Remarks to the Author):

Q: What are the noteworthy results?

A: See details below.

Q: Will the work be of significance to the field and related fields?

A: Yes, as a concept, however there are some limitations in the investigation.

Q: How does it compare to the established literature? If the work is not original, please provide relevant references.

A: See details below

Q: Does the work support the conclusions and claims, or is additional evidence needed?

A: Yes

Q: Are there any flaws in the data analysis, interpretation and conclusions? Do these prohibit publication or require revision?

A: No

Q: Is the methodology sound? Does the work meet the expected standards in your field?

A: Yes, however there are still some limitations. See details below.

Q: Is there enough detail provided in the methods for the work to be reproduced?

A: Some information is missing. See details below.

This is my review of the NCOMMS-23-24909 paper "On the importance of high-resolution and real-time impact forecasting for flood early warning" by Najafi et al. I feel reasonably well qualified to review this manuscript and do not have a conflict of interest with the authors, whereas my criticism solely aims to improve the quality of the paper.

In this paper, the authors established an impact-based flood forecasting system that is driven by a coupled hydrologic-hydrodynamic model using the mHM and RIM2D models respectively. Probabilistic meteorological forecasts are used to drive the hydrological model and predict different realisations of flood severity, inundation maps and consequently impacts on inundation depths and extents, flow

velocities, and affected people and infrastructure. The meteorological uncertainty, described through the ensemble set of 20 members, is the only source of uncertainty considered which is propagated to the impact assessment. The innovation of this assessment conducted in the Ahr Valley during the July 2021 flood event is based on the provision of high-resolution information (about 1km for hydrological forecasts and 10m for flood inundation mapping).

Overall, this article reads well and the presented material (figures and tables) supports the conclusions. I am aware of the work that the authors conduct which is very important for hydrological monitoring and operational flood early warning systems. I was lured by excitement by reading the title of the work; it was however unfortunate that despite the nicely written manuscript, I am lacking of seeing the key scientific breakthrough that would allow publication in prestigious journals like Nature Communications. To me, this manuscript is classified as a technical article, with keywords of the title also not being reflected in the analysis, i.e. the importance of impact-based FEWS is not numerically quantified due to lack of a benchmark.

I would propose the consideration of the manuscript to another more technical (scientific) journal, considering kindly the general comments that I provide below for improvements. Kindly note that my review does not aim to undervalue the work, but rather find the right audience for a technical note. To assist, I also provide a set of specific comments below.

Best regards,

General comments

1. As mentioned earlier, this investigation reads as a technical paper, without highlighting the need to evolve computational resources to accommodate such numerical developments. More and more national hydro-met services are investing to a paradigm shift from traditional FEWS to impact-based FEWS at high resolution. Accommodating such real-time services at the global scales is however very challenging (almost impossible) due to the trade-offs between computational power, scheduling of operational services and data storage archiving. Elaboration towards this direction is currently lacking.
2. The authors claim that uncertainty is considered in the impact-based flood forecasting. However, the only source of uncertainty considered here is the meteorological, which is also subject to the 20 ensemble members. Although this meteo system is state of the art for Germany, other prediction systems consist of double or even triple size of ensemble members. How could the proposed coupling of hydrological-hydrodynamic modelling address such high computational burdens? Regardless, the approach followed here is taking into account the median of the hydrological ensembles and also the min/max to represent the less and most severe impact of flooding. With this the hydrodynamic expense increases thrice in real-time. How come this method is revolutionary versus pre-calculated inundation maps that are re-called depending on these 3 water level values?
3. The title refers to the importance of high-resolution and real-time impact forecasting, however how is that importance quantified here? I cannot understand what is considered as a benchmark for this statement. Also it is important to note that this work addresses riverine flooding, and not other types of flooding such as flash floods.
4. The authors consider persistency as a forecast characteristic (3 consecutive initializations that exceed HQ100) to drive decision-making and flood notifications. However, there is no indication of forecast performance or skill as a property to condition the maximum lead time that forecasts can be trusted. This is quite important otherwise there will be a high increase of false alarms which can further result to lack of trust from the public/users. It would be important to provide information on the mHM model historical performance for the case study.
5. I would also argue that the definition of "lead-time", as it is given here, should be changed to "maximum lead-time". Lead-time is an indicator to define the future forecast time-step that we want

to assess.

Specific comments

Lines 33-34: "climate adaptation tools". Why do you associate a FEWS to climate adaptation? EWS are meant to guide decision-making that help mitigate flooding and its impacts caused by extreme weather events.

Lines 49-51: "Forecasting science... computational efficiency..." Are you referring to hydrological or meteorological or both types of forecasting? What about monitoring efforts that can be used in model parameterization and model initialization?

Line 50: "process representation and parameterization,"

Line 57: "Its first component...". Note that Figure 1 shows the observations as the first component.

Lines 68-69: This statement re hyper resolution (0.1-1km) is too ambitious to accept it based on the current hydrological tools. This has been achievable from a land surface modelling perspective but for sure not from a hydrological modelling perspective. The difference in spatial resolution between these two modelling types is large, as is also the performance and forecasting skill, which is very important in operational services. Kindly rewrite the statement and/or explicitly indicate land surface modelling here.

Line 72: "downscaling and initialization."

Line 78: "hydro-meteorological forecasts..."

Line 107: How large is the Ahr Valley?

Line 127: "considers different sources..."

Line 135: "ensemble predictions (16 initializations x 20 members)"

Lines 162-163: By how much is the predicted precipitation lower than the observed? And which were the reasons for predicting such low amounts?

Line 207: "impact forecasting for floods is possible..."

Line 212: "(see e.g. Apel et al. 38), is the..."

Line 226: "Here, the selection..."

Lines 226-227: This is the concept of forecast persistency, but how do you support this statement? Maybe add references.

Line 235: "shown in Figure 3(a)... indicate lead-time..."

Line 243: "the lead-time that forecasts exceed HQ100..."

Lines 242-244: This sentence is not right to me. It does not make sense though, because forecast persistency aims to filter hydrological signals from members that are driven by the chaotic meteorological input and hence avoid fault alarms. Otherwise we will have so many that result to an untrusty FEWS.

Line 255: "The near-real-time flood impact ..."

Lines 270-271: I believe that the run-time depends on the modelling time-step (here hourly), so kindly add this information in the sentence.

Lines 327-328: The last sentence is not needed here and is a repetition to what already exist in "Quantitative impact forecasting"

Lines 330-331: "buildings... roads... This information is..."

Line 334: "and calibration:"

Line 342: "forecast confidence:"

Line 343: "through the modelling"

Lines 347-348: "is defined as the maximum lead-time. The definition of maximum lead-time with..."

Reviewer #3 (Remarks to the Author):

Well done on a very good article. The authors present an advanced flood early warning system, demonstrating it's use using the 2021 European Summer floods and a case study in Germany. The article is very well written and detailed. I am not sure the title matches well with the content of the article. The content is quite technical and goes into detail about the approach. However, the title suggests a more high level overview of the utility of flood early warning systems. I also do not see where high- resolution is defined. I have provided some comments and suggestions below. My main recommendations is to change the title, and to comment how this work can be adapted to other areas, particularly those with less high quality data.

Comments:

- Well-written
- The 'Flood Early Warning Systems (FEWS): state-of-the-art, challenges, and future perspectives provides a really nice overview. It might be good to have an overview, possibly as a table, on what countries/authorities use for FEWS. This could help readers relate more to your example.

Major

- Title. In it's current guise it does not really match the topic of the article. Reading the current title, a reader would suggest you are writing about how important high resolution and real-time impact for forecasting are, rather than presenting a method/model demonstrating what is possible. Also what do you mean by high-resolution? I don't think you define it (only hyper resolution for hydrological models)
- Your example is a good one using the River Ahr. However, the sort of data required for this model is not available in most of the world and mostly where the deadliest floods occur. Many readers, especially being published in such a high profile journal, may think this system can be employed everywhere which of course is not the case. Therefore, it might be good to flag the data requirements, or even better to test with inferior, global data to see how results change. Then there is quite a simple argument – You need data of X quality to give useable results. Or you could just discuss.
- Related to the above, I think the statement on line 256 'The near-real-time impact forecasting study performed here is a proof-of-concept that can be implemented in any other region around the world with reasonable run times' may need some caveating. What data and computing power are needed? Are gauge data essential? Does a flood threshold need to be defined? Actually you do start to go into this in the next sentence, but could potentially do with more clarification.

Minor

- Line 30. You mention unprecedented rainfall events and associated flooding are more likely to occur. It would be good to have a discussion later in the article how these sort of events can be modelled in FEWs, especially given cognitive biases.
- Line 108 – exceeded might be more appropriate than crossed
- I really like using the Technology readiness level in Figure 1. Maybe you could provide a reference, or some indication what the numbers mean in the caption to help readers not familiar with it.
- Figure 2 – Do you think you could also put the 50 year return period to align with Rhineland-Palatine? Could be interesting.
- It would be good to provide a brief explanation of what a lead-time map is. It took me a while to get my head around it. Visually it is very initiative, but some explanation/definition would be helpful, especially to readers not so familiar with the technicalities.
- Table 2 caption. I think you are missing the word inundation in there. i.e total length of roads and railways inundated
- You could argue that future studies are needed to find out how your results/state-of-the-art FEWS can be best communicated.
- Figure 4 is difficult to read. Could you make it 1 column and 3 rows, thus increasing the size of each panel.

Response to Referee #1 comments

The paper titled "On the importance of high-resolution and real-time impact forecasting for flood early warning" by Najafi et al tackles a very important topic. However, I have major concerns about the focus and novelty of the paper.

We thank reviewer for taking the time to review our paper and for providing valuable feedback. We appreciate the reviewer's insights and have addressed all of their concerns. See below our detailed responses.

1 Comment 1

1.1 *The paper states in the abstract: "Here, we demonstrate that better informed crisis management and significant loss reduction can be achieved with a FEWS that includes extended components for probabilistic inundation and impact forecasting." However, the paper contains no such demonstration. The paper presents a modelling chain for ensemble forecasting of flood inundation and impacts. It does not present a framework on how the ensemble information can be used in decision-making during crisis management, achieving loss reduction, for example, how many lives might have been saved in the case study.*

1.1. We thank reviewer for their feedback and valuable insights. We acknowledge the concern that our paper does not directly demonstrate loss reduction or quantify lives saved, which can be challenging for a single event. A hazard causes losses only when vulnerable people and assets are exposed to it[1]. It is difficult to assess the number of lives which could have been potentially saved due to use of impact forecasting since we cannot replay the event. Such estimates can potentially be approached in long-term studies by comparing the situation prior to and after introducing impact-based forecasting.

Our abstract statement aimed to emphasize the potential of an enhanced Flood Early Warning System (FEWS) for early action and preparedness, but this may have inadvertently overstated the paper's content. Our paper primarily presents a modeling chain for ensemble forecasting of flood inundation and impacts. While this advancement is significant, we recognize the need for a more explicit discussion on how it can support informed decision-making.

To address this, we would like to reference to Zurich Insurance report of the event[2]. It highlights the challenges faced by local authorities and first responders due to their lack of apprehension about forecasted heavy rainfall and water levels. For instance, understanding what 150 mm or 200 mm of rain or a gauge level of 6 m might mean on site, and what appropriate preparation and response measures should be taken. This report underscores the need to better understanding of practical implications of such forecasts and the corresponding preparation and response measures. The abstract and manuscript are revisited to ensure readers are not misled, and to represent the research accurately:

1.1.1. Revisions in the abstract

To address the reviewer's comment, we revised the sentence in the abstract as follows:

"High-resolution, impact-based flood forecasts provide insightful information for better-informed decisions and tailored emergency actions."

We would like to elaborate on the usage of the word "better" by comparing what we have proposed in the proposed forecasting chain and its comparison with the official forecast by the authority and EFAS.

EFAS: For catchments $< 2000 \text{ km}^2$, EFAS calculates flash flood indicators at so called ERIC reporting points and do not represent water level or discharge forecasts on gauge data[3]. These are mainly based on maximum accumulated surface runoff and updated weather forecasts (ensemble forecasts COSMO-LEPS). The EFAS flash flood forecasts are updated only twice a day, use relatively coarse spatial resolution weather forecasts (grid cell COSMO: 49 km^2).

The results are classified according to the following return period ranges: > 2 -year flood, > 5 -year flood, and > 20 -year flood, where the statistics are again not based on gauge measurement time series but on 20-year model simulations.

Official forecast by LfU: This official water level forecasts are updated at least eight times a day, and are based on more up-to-date, spatially resolved weather forecasts (grid cell ICON-D2: 4 km^2), the current measurement data of numerous measuring stations (precipitation and gauges), and spatially and temporally high-resolution water balance models that have been explicitly calibrated for high-water-relevant gauges.

Neither of those flood forecasting systems provides impact-based forecasting based on hydrodynamic inundation modeling at spatial resolution similar to one presented in our manuscript, which is a significant added value. To this end, it is demonstrated that better informed disaster management can be achieved with a FEWS that explicitly includes extended components for inundation modeling and impact forecasting.

1.1.2. Revisions in the main manuscript

With the availability of probabilistic impact forecasts, decision makers should act based on a protocol of action or self-determined[4] probability thresholds. The protocol of action may vary based on the specific hazard, the region, and the institutional arrangements in place, institutional capacities, and cultural factors. Elaborating on these protocols is out of scope of this manuscript. However, we have touched upon some of these aspects in a newly added section entitled "Enhancing Flood Forecast Communication for Informed Decision-Making and Risk Management" to the revised manuscript (L388-ff). In this section, theoretical models and decision analysis methods for decision making under uncertainty such as Protective Action Decision Model (PADM)[5] and cumulative prospect theory [6] are referred (L395-ff).

The new information in this section has shown that, the ensemble median of forecasted inundated area consistently exceed threshold of flood hazard mapping for all forecasts issued in a lead-time of 17 hours to the flood peak. In addition, convergence of forecasts from 8 to 2 hours prior to flood can indicate an imminent disaster. Such a-priori information are crucial for well-informed decision-making during crisis management, and possibly leading to saving of lives. Moreover, we highlight the importance of effective communication of probabilistic impact-based warning in lines 463-467.

1.2 *In the case study, the maximum forecast ensemble member was shown to better align*

with the final observation than the median forecast. In general, however, how should the decision-makers use the full ensemble distribution? In another event, the maximum forecast member may turn out to be much too high compared with observation. Therefore, using the highest forecast member to guide decision-making may not be a good approach.

In response to the reviewer's concern, we would like to clarify that nowhere in the manuscript did we suggest using the highest forecast member as a guide for decision-making. As stated in point 1.1, the primary focus of this study is to provide proof that a skilful impact-based forecast and warning could have been issued with enough lead time to reduce the risk within the proposed forecasting chain. Presenting expert information should enable the creation of a convincing warning narrative that ultimately supports decision-makers and encourages action. By providing the most likely, the probable worst case and the probability of exceeding a particular warning threshold, we have shown that the resulting probability distribution from ensemble forecasting should then be interpreted. What our analysis emphasises is the use of a probabilistic approach which is flexible enough to provide information for any probability of interest that can be adapted for any other percentiles that decision-makers would like consider (e.g., terciles, quantiles, 90% confidence etc . . .). Information on these flood impact relevant indicators with different lead times can support decision-makers and allow identification of what will be impacted, where, when and to which extent.

Furthermore, we emphasise that the tasks and duties of decision makers and responding agencies are strongly influenced by national legislation and vary between countries. Decision makers should therefore decide according to the protocol of action based on country's specific legislation. Ideally, such guidelines/frameworks should specify how to use ensemble forecasting based information for decision-making. Developing such a guideline requires a participatory approach to identify how to communicate probabilistic information in interdisciplinary research groups, and how to develop ensemble-based, user-optimised forecast products. This is, however, outside the scope of this paper.

Here, we would like to address a study which has shown the key criteria for communicating probabilistic information and integrating it into decision processes [4]. The study showed no single probability threshold for issuing a warning that could account for every user's needs - particularly in large user groups. The decision-making that goes into assessing risk inevitably requires a series of judgements to be made, to refine the circumstances or scope of the situation[1]. Interpretations of probabilities also differ between experts and non-experts, and even within expert groups [7]. It means that decision-making authorities should adjust and continuously adapt the thresholds based on their day-to-day experience without an upper level document considered as a protocol of action. The World Meteorological Organisation (WMO) has published several guidelines on communicating forecast uncertainty, which can be a good basis for decision makers.

To summarize, no "one-size-fits-all" solution or a general approach could be suggested on the question of how the decision-makers should use the information on full ensemble distribution. The summary of this response is provided in lines 412-ff at the end of the newly added section "Enhancing Flood Forecast Communication for Informed Decision-Making and Risk Management" in the revised manuscript.

- 1.3 *On a minor point, equating the largest member from an ensemble of size 20 to 5% chance of exceedance can be misleading as the largest member will be subject to sampling error. One would need a much larger ensemble size to estimate the 95% quantile.*

We agree with reviewer that the ensemble size of 20 is not large enough to estimate 95% quantile. Referring to the probability of 5 percent was therefore removed from the corresponding sentence.

2 Comment 2

2.1 *A prototype of flood forecasting modelling chain that includes ensemble/probabilistic flood inundation forecasting is not new. For example,*

- Gomez et al (2019) Skill of ensemble flood inundation forecasts at short- to medium-range timescales, Journal Hydrology, <https://doi.org/10.1016/j.jhydrol.2018.10.063>
- Schumann et al (2013) A first large-scale flood inundation forecasting model, Water Resources Research, 0043-1397/13/10.1002/wrcr.20521
- Ivanov et al (2021) Breaking Down the Computational Barriers to Real-Time Urban Flood Forecasting, Geophysical Research Letters, <https://doi.org/10.1029/2021GL093585>

I do, however, acknowledge that ensemble flood impact forecasting is relatively new.

We thank reviewer for pointing out the existing works on flood forecasting modeling chains that include ensemble/probabilistic flood inundation forecasting. We acknowledge these contributions in lines 134-135. Table 1 in this response, summarizes the key features and outcomes of these studies to offer a clear comparison with our work. Furthermore, we highlighted the aspects where our study brings novel contributions to the field of ensemble flood impact forecasting in lines 166-ff. In particular, we have emphasized our methodological advancements, and how we overcome the computational barriers for real-time urban flood forecasting.

Table 1: Summary of different studies

Reference/ study	Hydrologic model set-up	Hydrologic model Res	Hydrodynamic model Res	Hydrodynamic variable	Ensemble No.
Gomez et al. 2019	Hourly	4 km	1 m	1D (flow discharge)	11
Schumann et al. 2013	Daily	0.25 - degree	1 km	2D (flood inundation)	50
Ivanov et al. 2021	Hourly	(Surrogate)	(10 ⁰ - 10 ¹) m	(Surrogate)	10000 rainfall realizations
Najafi et al. (in review)	Hourly	1 km	10 m	2D (flood inundation)	20

2.2 *I question several critical statements made in the paper:*

“The near-real-time impact forecasting study performed here is a proof-of-concept that can be implemented in any other region around the world with reasonable run times.”

What is the basis for this generalisation?

We thank reviewer for insightful comment. This sentence has been removed. Instead, a sub-section is provided in the Method section (lines 553-ff) which presents computational resources and data requirements to develop a similar system.

It is widely known that the large computational demand for running high-resolution hydrodynamic models poses a real barrier for producing ensemble flood inundation forecasts. The authors have not explained how they solved the problem by using the hydrodynamic model RIM2D.

The computational performance of RIM2D has been addressed By Apel etl., (2022)[8]. Fast hydrodynamic RIM2D simulations have been achieved mainly due to its massive parallelization on graphical processor units (GPUs)[8]. In this respect, ensemble forecasts pose no disproportional computational burden, as they can be run in parallel on separate GPU devices. With the adopted computational approach, we are able to process a few dozen ensemble runs for a single event even on a medium class GPU server. These details are added in the revised manuscript in lines 559-ff.

Is RIM2D much faster than other hydrodynamic models? What is the computational requirement of RIM2D for a range of catchments/floodplains?

We would like to refer to RIM2D simulation for Ahr published by Apel et al., (2022): "The 30 h long flood event was simulated in 14 minutes on an NVIDIA Tesla P100 GPU computing unit connected to a Linux server with an Intel Xeon Gold 6140 CPU. This is a simulation runtime equivalent to less than 0.8% of the simulated event duration. The memory capacity of the GPU unit in terms of computational nodes was used to about 15% only, leaving room for increasing the model domain or spatial resolution."

In the same reference[8], it has been addressed that "RIM2D implements the same numerical core as the model LISFLOOD-FP but coded in CUDA Fortran and implemented to run on large NVIDIA Tesla graphical processor units (GPUs). This implementation enables massive parallelization of the numerical computations at low costs compared to large multi-core computing clusters". RIM2D run-time scales linearly with the domain size.

"Extending flood forecasts from streamflow and water levels at single river gauges to spatially distributed information on inundation, flow velocities and further impacts has been considered unfeasible for many years because of the extensive runtime of hydrodynamic models for an ensemble forecast in real-time and the lack of river cross-section data at a reasonably high resolution." As my comment in the last paragraph, the authors have not explained how they overcome the problem.

We would like highlight that in the study by Apel et al., (2022), some simplified assumptions were made in the RIM2D model setup, including the non-consideration of the detailed river bed bathymetry. River channel geometry is considered as far it is contained in the DEM. Particularly for large floods with overbank flow, where the performance of FEWS gains importance, the role of the channel bathymetry diminishes. In the mentioned study, the authors have shown the validity of the simplified assumptions in the model setup supporting the model performance[8].

"The RIM2D simulation, with its short run-time of approximately 22 minutes for a 48-hour model simulation, ...". The authors state that 20 ensemble members are available every 3 hours. With a run time of 22 min, they cannot be run after each other on a single computer, as this would take longer than 3 hr. The authors have not described how they will set up a system that can run all 20 ensembles every 3 hours in real-time. What computers and how many are needed?

We thank reviewer for raising this important question. We apologize if there was any lack of clarity in the initial explanation and appreciate the opportunity to elaborate. Each simulation ensemble member is independent and can be established on a single GPU device. This design facilitates parallel processing where multiple simulations can be executed simultaneously, ensuring that the ensemble forecasts are generated within the stipulated 3-hour window. On a computational server equipped with multiple GPU devices, some or even all ensemble members can be run simultaneously. In our original manuscript, we should have emphasized more adequately that we don't necessarily run all 20 ensemble members every 3 hours. We prioritize calculating the ensemble minimum, median, and maximum, which provides insight into the entire range and central tendency of forecasted flood impacts. In the revised version of the manuscript, we have expanded these details and included the 25th and 75th percentiles to enrich the decision-making information without considerably increasing the computational load. The enhanced clarification in the revised manuscript is provided below:

"The RIM2D simulations are executed on the Graphical Processor Units (GPUs) to achieve high computational performance. Each ensemble run is allocated to a single GPU device allowing for parallel processing. While 20 ensemble members are available, our real-time forecasting focuses on selected percentiles with respect to peak discharge at the upstream boundary (minimum, 25%, median, 75%, and maximum). This approach ensures timely forecasts every 3 hours and is able to accommodate larger ensembles if needed."

"While running a full ensemble-based hydrodynamic modeling runs in near real-time can still be limited with respect to ensemble size and catchment areas, it can provide information on uncertainty and probability of streamflow/water level exceeding critical thresholds

for early warning". The logic in this sentence is unclear to me.

We thank reviewer for the comment. We have reviewed the sentence in question and agree that it may not have been clearly articulated. To enhance the clarity and focus of our manuscript, we have decided to remove this sentence. The changes have been made in lines 177-ff of the revised manuscript.

2.3 I should mention that there is active research in the development of fast emulators of high-resolution hydrodynamic models, for the purpose of ensemble flood inundation and impact forecasting.

Thanks for the comment. This has been addressed in lines 144-ff.

3 Comment 3 (minor)

In general, the writing needs much improvement, especially on the balance of content. There is not enough material/emphasis on the "new" work, that is, ensemble flood inundation and impact forecasting (and the use in decision-making). The conclusions have a mix of general conclusions and conclusions specific to the case study; it is not always clearly described.

We thank reviewer for the comment. We addressed this by discussing existing approaches and their challenges. In addition, a new section is added to represent how our suggested approach can support tailored decision making (lines 390-ff). The discussion section is also revised and it does not address specific conclusions to the case study.

References

- [1] Golding, B. *Towards the "Perfect" weather warning: bridging disciplinary gaps through partnership and communication* (Springer Nature, 2022).
- [2] M., S. *et al.* PERC floods following "Bernd". Tech. Rep., Zurich Insurance Company (2021).
- [3] LfU. Hochwasser im Juli 2021. Tech. Rep., Landesamt für Umwelt (LfU) Rheinland-Pfalz (2022). URL https://lfu.rlp.de/fileadmin/lfu/Wasserwirtschaft/Ahr-Katastrophe/Hochwasser_im_Juli2021.pdf.
- [4] Fundel, V. J., Fleischhut, N., Herzog, S. M., Göber, M. & Hagedorn, R. Promoting the use of probabilistic weather forecasts through a dialogue between scientists, developers and end-users. *Quarterly Journal of the Royal Meteorological Society* **145**, 210–231 (2019).
- [5] Lindell, M. K. & Perry, R. W. The protective action decision model: Theoretical modifications and additional evidence. *Risk Analysis: An International Journal* **32**, 616–632 (2012).
- [6] Tversky, A. & Kahneman, D. Advances in prospect theory: Cumulative representation of uncertainty. *Journal of Risk and Uncertainty* **5**, 297–323 (1992).
- [7] Kox, T., Gerhold, L. & Ulbrich, U. Perception and use of uncertainty in severe weather warnings by emergency services in Germany. *Atmospheric Research* **158**, 292–301 (2015).
- [8] Apel, H., Vorogushyn, S. & Merz, B. Brief communication: Impact forecasting could substantially improve the emergency management of deadly floods: case study July 2021 floods in Germany. *Natural Hazards and Earth System Sciences* **22**, 3005–3014 (2022).

Response to Referee #2 comments

Q: *What are the noteworthy results* A: See details below.

Q: Will the work be of significance to the field and related fields? A: Yes, as a concept, however there are some limitations in the investigation.

Q: How does it compare to the established literature? If the work is not original, please provide relevant references. A: See details below

Q: Does the work support the conclusions and claims, or is additional evidence needed? A: Yes

Q: Are there any flaws in the data analysis, interpretation and conclusions? Do these prohibit publication or require revision? A: No

Q: Is the methodology sound? Does the work meet the expected standards in your field? A: Yes, however there are still some limitations. See details below.

Q: Is there enough detail provided in the methods for the work to be reproduced? A: Some information is missing. See details below.

This is my review of the NCOMMS-23-24909 paper "On the importance of high-resolution and real-time impact forecasting for flood early warning" by Najafi et al. I feel reasonably well qualified to review this manuscript and do not have a conflict of interest with the authors, whereas my criticism solely aims to improve the quality of the paper.

In this paper, the authors established an impact-based flood forecasting system that is driven by a coupled hydrologic-hydrodynamic model using the mHM and RIM2D models respectively. Probabilistic meteorological forecasts are used to drive the hydrological model and predict different realisations of flood severity, inundation maps and consequently impacts on inundation depths and extents, flow velocities, and affected people and infrastructure. The meteorological uncertainty, described through the ensemble set of 20 members, is the only source of uncertainty considered which is propagated to the impact assessment. The innovation of this assessment conducted in the Ahr Valley during the July 2021 flood event is based on the provision of high-resolution information (about 1km for hydrological forecasts and 10m for flood inundation mapping).

Overall, this article reads well and the presented material (figures and tables) supports the conclusions. I am aware of the work that the authors conduct which is very important for hydrological monitoring and operational flood early warning systems. I was lured by excitement by reading the title of the work; it was however unfortunate that despite the nicely written manuscript, I am lacking of seeing the key scientific breakthrough that would allow publication in prestigious journals like Nature Communications. To me, this manuscript is classified as a technical article, with keywords of the title also not being reflected in the analysis, i.e. the importance of impact-based FEWS is not numerically quantified due to lack of a benchmark.

I would propose the consideration of the manuscript to another more technical (scientific) journal, considering kindly the general comments that I provide below for improvements. Kindly note that my review does not aim to undervalue the work, but rather find the right audience for a technical note. To assist, I also provide a set of specific comments below.

Best regards,

Response to Referee #2 comments

We thank the reviewer for taking the time to review our manuscript and detailed feedback. We appreciate the reviewer's expertise in the subject and are grateful for their objective and constructive criticism aimed at enhancing the quality of our work. We value the reviewer's insights in this process and are pleased that the reviewer recognized and acknowledged the conceptual importance of our work. The reviewer's feedback about the limitations in our methodology is noted and is fully taken into account in the revised manuscript to enhance and highlight its unique contributions. The reviewer's point that some information is missing, is also taken seriously, and we have ensured to provide a more comprehensive description in the revised manuscript.

We understand the reviewer's feedback regarding the need for numerical quantification for the im-

portance of impact-based FEWS. The result and methods used in the original submission have been compared to several benchmarks, including the official hydrologic forecasts at gauge, 75 high water marks at buildings reported by the inhabitants and flood extent mapped by the responsible authority (see [1]). It lacked a benchmark for impact-based forecasting as stated correctly by the reviewer. In the revised manuscript, however, we have included an explicit benchmark for impact-forecasting to give context to our results and show how our methodologies and findings can be compared to existing standards. This benchmark is added to underscore our findings' scientific validity and significance, further address the reviewer's concerns, and strengthen the work.

The reviewer has considered "high-resolution forecast for hydrological and inundation mapping" as a novelty of the work. We believe that providing probabilistic inundation and maximum lead-time maps, impact forecasting at the scale of residential buildings and transportational infrastructures, together with provision of information which can support tailored decision making should also be considered as innovative aspects of our work.

With respect to the scientific breakthrough point brought by the reviewer, we conducted an original work compared to established literature to provide a proof-of-concept for developing high-resolution flood impact forecasting system with the capability to provide forecasts in near-real-time. We showcase the possibility of generating high-resolution information and maps supporting operational decision-making based on state-of-the-art modeling and computational resources. Valuable information could be provided to local authorities to support hydrological monitoring and operational flood early warning systems. Freely available high-resolution weather forecasts, open source state-of-the-art hydrologic-hydraulic models, high-resolution digital elevation map, and open data with global coverage (open street map) can be used for informed decision making in real-time.

We truly appreciate the reviewer's general and specific comments. These insights are invaluable for improving the quality and clarity of our manuscript. We assure you that we will carefully consider each point in the revision. Please allow us to address each point the reviewer has raised in the following sections.

General comments

- 1 As mentioned earlier, this investigation reads as a technical paper, without highlighting the need to evolve computational resources to accommodate such numerical developments. More and more national hydro-met services are investing to a paradigm shift from traditional FEWS to impact-based FEWS at high resolution. Accommodating such real-time services at the global scales is however very challenging (almost impossible) due to the trade-offs between computational power, scheduling of operational services and data storage archiving. Elaboration towards this direction is currently lacking.*

We appreciate the comment from the reviewer and have considered this comment in the revision in the discussion section (lines 434-ff). We would like to emphasise that at no point did we assert that current FEWS are equipped to support real-time, impact-based services on a global scale. Our claim is specifically constrained to the regional scale, which we have supported with substantial evidence and analysis in our study.

Concerning the scheduling of operational services, the incorporation of workflow managers such as ecFlow [2], coupled with an intuitive graphical interface, streamlines the operational service schedule. This not only improves user engagement and accessibility but also contributes to optimizing service delivery in real-time hydrodynamic modelling and forecasting (refer to lines 447-ff). Furthermore, hydrodynamic modelling is initiated only when predetermined thresholds or user/decision-maker-defined exceedance levels are met. This automated triggering mechanism enhances system responsiveness and adaptability in real-time scenarios, thereby mitigating the need for extensive data storage archiving. We trust this clarification addresses the reviewer's concerns and underscores the contributions and innovations presented in our study.

- 2 The authors claim that uncertainty is considered in the impact-based flood forecasting. However, the only source of uncertainty considered here is the meteorological, which is also subject to the 20 ensemble members. Although this meteo system is state of the art for Germany, other*

prediction systems consist of double or even triple size of ensemble members. How could the proposed coupling of hydrological-hydrodynamic modelling address such high computational burdens? Regardless, the approach followed here is taking into account the median of the hydrological ensembles and also the min/max to represent the less and most severe impact of flooding. With this the hydrodynamic expense increases thrice in real-time. How come this method is revolutionary versus pre-calculated inundation maps that are re-called depending on these 3 water level values?

We thank the reviewer for their feedback. We have addressed each of their relevant points as follows.

Source of uncertainty *We acknowledge the existence of other NWP and consider rainfall uncertainty from NWP through a more extensive ensemble set. Since March 2020, we have been privileged to have access to real-time forecasts from ECMWF and our initial approach to flood forecasting employed the ECMWF Medium-range forecasts (Integrated Forecasting System (IFS)) system wherein mHM was forced by 51 real-time ensemble members derived from IFS (IFS-mHM). We present these results in Figure 1 of this response.*

The flood forecast peaks shown in Figure 1 are clearly underestimated. By comparing the results from two different NWP, we realized that ICON_D2_EPS-mHM modeling chain improves the flood forecasting system with the compromise of less ensemble members. Therefore, we decided to consider a high-resolution NWP with spatial resolution of 2.2 km which is significantly higher than horizontal grid resolution of IFS (18 km). This enhanced resolution facilitates more accurate predictions to forecast convective events, crucial for effective flood management.

The German Weather Service (DWD) and official flood forecasting systems in Germany also use the same NWP for issuing early warnings. However, the selection of ICON_D2_EPS was based on more than just the horizontal resolution but also on the frequency of forecast initialisations. This system provides more frequent forecasts (eight forecasts per day) than IFS making it very suitable for flood forecasting. The utilisation of 320 ensemble members over a 48-hour lead-time offers a comprehensive view of potential meteorological developments. That said, the uncertainty from atmospheric boundary conditions was actually investigated by the authors. The results from ECMWF_IFS-mHM system were presented in an earlier work by Najafi et al. (2022) at the AGU 2022 Fall Meeting[3].

Hydrological model structural uncertainty was not considered in this study since mHM has shown good performance in reconstructing and simulating high flows and flood peaks during the historical period, also in comparison to other well-established hydrological models over Europe and the U.S., see, e.g., [4, 5]. Furthermore, one can also refer to the various studies indicating that parameter uncertainty is generally of lower magnitude with respect to uncertainties for rainfall forecast, e.g., [6].

Computational Burden with Larger Ensemble Sizes *Our approach to utilizing selected percentiles is highly scalable and adaptable. We designed it to seamlessly integrate ensemble hydrological forecasts of varied magnitudes. The number of hydrodynamic model runs would be similar for selected percentiles even for NWP with double or even triple the ensemble members. As stated in our previous comment, the flexibility and efficiency of this methodology ensure that uncertainty from atmospheric initial conditions is quantified. The only limitation here would be the number of available GPUs in the server to support additional percentiles of interest for end-users.*

Median, Min/Max of Hydrological Ensembles *To better communicate the forecast uncertainty, we have provided additional uncertainty ranges corresponding to percentiles of 25% and 75% to show the flexibility of the system; and we have included a new figure (Figure 4) in the manuscript addressing how whisker plots can be used for informed decision making (please see lines 340-343), Table 2 and the new section entitled "Enhancing Flood Forecast Communica-*

Comparison with Pre-Calculated Inundation Maps We acknowledge the use of pre-simulated scenarios in flood forecasting, which could minimize the computation time of a fully operational framework. Yet, the methodology proposed in our study offers several notable advantages that augment the efficiency and accuracy of flood forecasting. Our approach champions the use of real-time forecasts as a supplementary informational layer, enriching pre-calculated hazard maps and facilitating the incorporation of prevailing antecedent conditions (Speight et al. 2020). We consider the proposed approach based on the dynamic physically-based simulation of real on-setting flood events to be superior to a combination of pre-calculated flood maps for several reasons.

1. *Incorporation of Real-Time Flood Dynamics:* Pre-calculated inundation maps often need to account for flood events' fluid, dynamic nature[1]. Flood events exceeding the scenarios used to develop pre-calculated maps cannot be reliably extrapolated. This was also the problem in the study region of the Ahr basin, where the previously calculated flood outlines for very extreme events were exceeded by far[7]. Dynamic simulation therefore, offers more reliable and accurate inundation maps compared to pre-calculated maps.
 2. *Quality Assurance and Comprehensive Event Library required for pre-calculated hazard maps:* Pre-calculated hazard maps typically hinge on the assumption of a seamless connection between real-time forecasting models and static inundation and impact assessments[8]. The efficacy of this approach is contingent upon several factors: (1) the precision of the original hydraulic inundation modeling, (2) the accurate depiction of urban drainage capacity, and (3) a comprehensive library of events, encompassing diverse return periods and event duration, ensuring a nuanced response to forecasted rainfall events. These assumptions[8] might not be valid for all flood events. Furthermore, pre-calculated maps are based on the assumptions of specific typical flood flow hydrographs. Substantial inaccuracies may arise in cases, where flood hydrographs show a different form e.g. double-peak or larger flood volumes if generated by processes other than for typical events. The pre-calculated maps have inconsistencies across their borders since they are not resulting from a spatially consistent simulation of one event, but from different events driven by spatially inconsistent flood hydrographs at different gauges.
 3. *Dynamic simulation can provide invaluable information related to the temporal evolution of an event, such as e.g. the time left for achieving a certain water level or the rate of water rise. These indicators can be decisive for successful evacuation and emergency measures.*
 4. *Adaptive Resolution of Flood Hazard Maps:* Our method introduces an enhanced adaptability in the resolution of flood hazard maps, mainly when high-resolution Digital Elevation Models (DEMs) are accessible. It supports the integration of low-fidelity models while maintaining acceptable error margins, offering a blend of accuracy and computational efficiency.
- 3 *The title refers to the importance of high-resolution and real-time impact forecasting, however how is that importance quantified here? I cannot understand what is considered as a benchmark for this statement. Also it is important to note that this work addresses riverine flooding, and not other types of flooding such as flash floods.*

We appreciate the reviewer's comments and queries. We fully understand the reviewer's concerns; therefore, we have revised the title and addressed each point to provide clarity.

Quantifying the importance of high-resolution and real-time impact forecasting: *The proposed study provides a real-time forecasting chain at the scale of individual buildings with a spatial resolution of 10 m. Providing flood impact forecasts at this object scale would only be possible by using high-resolution forecasts[9].*

To address the review's concern, we add a satellite-based damage assessment in the revised manuscript as a benchmark for impact forecasting, the Copernicus Emergency Management Service On Demand Mapping. This product (shown in Figure 3) was triggered by the German

Joint Information and Situation Centre (GMLZ) to monitor the flood evolution. It includes an assessment of the event's impact and extent derived from images acquired as soon as possible after the emergency event. Details relevant to the Copernicus Emergency Management Service is added to the Method section of the revised manuscript in lines (515-ff). The results of the comparison are provided in Table 2 of the revised manuscript.

Benchmark for Statement: *The basis of the real-time impact forecasting is hydrodynamic modelling in the proposed forecasting chain. Its evaluation has been provided in the reference Apel et al., (2022)[1]. The results showed a high agreement between our hydrodynamic simulations and the post-event mapping of the inundated areas and respective estimates given by the official authority of Rhineland-Palatinate (State Office for the Environment; LfU), supporting confidence in the simulation results. In addition to inundation extent, water depths from 75 high water marks (red dots in Fig. 2b of [1]) were used for the evaluation. The mean bias over all marks is -0.39 m with an RMSE of 0.66 m.*

Focus on Riverine Flooding: *This concern has been addressed in the revised title. Additionally, our model's efficacy is assessed through benchmarking and comparison between post-event mapping of the inundated areas by the official authority, water marks and geo-spatial information satellite images. We hope this addresses the reviewer's concerns adequately and provides a clear insight into our approach, its benchmarking, and its specialised focus.*

- 4 *The authors consider persistency as a forecast characteristic (3 consecutive initialisations that exceed HQ100) to drive decision-making and flood notifications. However, there is no indication of forecast performance or skill as a property to condition the maximum lead time that forecasts can be trusted. This is quite important otherwise there will be a high increase of false alarms which can further result to lack of trust from the public/users. It would be important to provide information on the mHM model historical performance for the case study.*

We appreciate the reviewer's insightful observation regarding the incorporation of persistency as a forecast characteristic without an explicit indication of the forecast performance or skill. We also agree that providing the forecast performance or skill is essential for end users. We tried to clarify this point for the illustrated case of our work.

- 4.1. **No indication of forecast performance or skill as a property to condition the maximum lead time:** We have indeed employed persistency as a pivotal forecast characteristic, similar to established flood forecasting systems like EFAS [10]. Extreme events, such as the unprecedented 2021 summer flood, characterized by return periods exceeding a century, underscore the inherent challenges in forecasting very rare floods of this magnitude. However, the ICON_D2_EPS has been operational by the German Weather Service (DWD) since February 2021. The relatively brief operational period poses challenges in deriving comprehensive insights into its performance, particularly concerning false alarm ratios.

We concede to the criticality of evaluating false alarms as a metric in assessing the robustness of recently operational NWP, especially those without extensive hindcast data. However, we advocate that hindcast evaluations, even of isolated events as detailed in our study, can offer invaluable insights and to guide local authorities in anticipatory actions during analogous future events. In response to the reviewer's valuable feedback, we have enriched the discussion section of our manuscript in lines 449-453 of discussion to communicate the above-mentioned limitation. With that, we clarify that one limitation of the approach we have taken is lack of the possibility of system performance based on a long-term skill evaluation.

- 4.2. **mHM historical performance for the case study:** The mHM historical performance has been provided as an additional supplementary figure in the manuscript (see Figure 2 in this response. The figure is also added as Supplementary Figure S5) for the reference. The calibration is done for the period 2011-2020, which also coincides with the calibration period used in this study. Note that the second largest flood in Ahr river occurred in 2016 with a 100-year return period. Despite

the significant peak in 2016 being missed, the other major winter flood peaks were properly captured. Very good model performance is demonstrated by skill scores like KGE, which are consistently high (equal or above 0.9) across all three temporal aggregations.

5 I would also argue that the definition of "lead-time", as it is given here, should be changed to "maximum lead-time". Lead-time is an indicator to define the future forecast time-step that we want to assess.

We thank the reviewer for this insightful feedback. We appreciate the reviewer's attention to detail and their suggestion on the definition of "lead-time". We agree to change our definition from "lead-time" to "maximum lead-time" in the revised manuscript. This adjustment will better convey the intended meaning.

Figure 1: Flood forecasting in Ahr based on ECMWF_IFS-mHM system using 51 ensemble members.

Specific comments

1 Lines 33-34: "climate adaptation tools". Why do you associate a FEWS to climate adaptation? EWS are meant to guide decision-making that help mitigate flooding and its impacts caused by extreme weather events.

Thanks for the comment. We have revised the related text and modify "adaptation" to "crucial".

2 Lines 49-51: "Forecasting science . . . computational efficiency . . ." Are you referring to hydrological or meteorological or both types of forecasting? What about monitoring efforts that can be used in model parameterization and model initialization?

output_calib_02

Figure 2: Time series of streamflow simulations and observation at (a) hourly, (b) daily, (c) monthly time step. Monthly climatology is show in panel (d) and daily flow direction curve is shown in panel (e).

Here, we are referring to both. The reviewer's comment is applied in lines 55-ff of the revised manuscript.

3 *Line 50: "process representation and parameterization,"*

Thanks, it is revised based on the reviewer's comment.

4 *Line 57: "Its first component ...". Note that Figure 1 shows the observations as the first component.*

Thanks, it is revised based on the reviewer's comment.

5 *Lines 68-69: This statement re hyper resolution (0.1-1km) is too ambitious to accept it based on the current hydrological tools. This has been achievable from a land surface modelling perspective but for sure not from a hydrological modelling perspective. The difference in spatial resolution between these two modelling types is large, as is also the performance and forecasting skill, which is very important in operational services. Kindly rewrite the statement and/or explicitly indicate land surface modelling here.*

Thanks, we have revised the statement following the reviewer's suggestion in lines 86-ff.

6 *Line 72: "downscaling and initialization."*

Thanks, it is revised based on the reviewer's comment.

7 *Line 78: "hydro-meteorological forecasts ..."*

Thanks, it is revised based on the reviewer's comment.

8 *Line 107: How large is the Ahr Valley?*

The catchment area is 746 km². This information is now provided in the revised manuscript in line 212.

9 *Line 127: "considers different sources ..."*

Thanks, it is revised based on the reviewer's comment.

10 *Line 135: "ensemble predictions (16 initializations x 20 members)"*

Thanks, it is revised based on the reviewer's comment.

11 *Lines 162-163: By how much is the predicted precipitation lower than the observed? And which were the reasons for predicting such low amounts?*

For the 12-h target period of 07/14 10:00 to 07/14 22:00 CET, the difference between the quantitative estimate of precipitation and ensemble medians (for different forecast initialisations) varies between 10 to 80 mm (in the time window between 47 to 17 hour to flood peak). The strongest precipitation in the Ahr catchment area occurred in the afternoon and evening of 14th July; in connection with intensified convergence and uplift processes, which were influenced by the development of a marginal low pressure area. The prediction of the location and the track of this small-scale low-pressure area has a decisive impact on the location and time of the highest predicted rainfall amounts by the weather models. The expected precipitation amounts were difficult to localise by the weather forecasts some hours before the event. The high-resolution and convection-resolving model ICON-D2 locates the highest precipitation amounts on 14th July between 14:00 and 20:00 CEST very differently depending on the forecast time. We have provided the uncertainty associated with precipitation forecast for different initialisations and its quantitative estimation in Figure 2 of the revised manuscript.

12 *Line 207: "impact forecasting for floods is possible ..."*

Thanks, it is revised based on the reviewer's comment.

13 *Line 212: "(see e.g. Apel et al. 38), is the ..."*

Thanks, it is revised based on the reviewer's comment.

- 14 *Line 226: "Here, the selection ..."*
Thanks, it is revised based on the reviewer's comment.
- 15 *Lines 226-227: This is the concept of forecast persistency, but how do you support this statement? Maybe add references.*
A reference is added as suggested in support to this statement.
- 16 *Line 235: "shown in Figure 3(a) ... indicate lead-time ..."*
Thanks, it is revised based on the reviewer's comment.
- 17 *Line 243: "the lead-time that forecasts exceed HQ100 ..."*
Thanks, it is revised based on the reviewer's comment.
- 18 *Lines 242-244: This sentence is not right to me. It does not make sense though, because forecast persistency aims to filter hydrological signals from members that are driven by the chaotic meteorological input and hence avoid fault alarms. Otherwise we will have so many that result to an untrusty FEWS.*
We revised the sentence in line 362-ff.
- 19 *Line 255: "The near-real-time flood impact ..."*
Thanks, it is revised based on the reviewer's comment.
- 20 *Lines 270-271: I believe that the run-time depends on the modelling time-step (here hourly), so kindly add this information in the sentence.*
Thanks, it is revised based on the reviewer's comment.
- 21 *Lines 327-328: The last sentence is not needed here and is a repetition to what already exist in "Quantitative impact forecasting"*
Thanks, it is revised based on the reviewer's comment.
- 22 *Lines 330-331: "buildings... roads... This information is ..."*
Thanks, it is revised based on the reviewer's comment.
- 23 *Line 334: "and calibration:"*
Thanks, it is revised based on the reviewer's comment.
- 24 *Line 342: "forecast confidence:"*
Thanks, it is revised based on the reviewer's comment.
- 25 *Line 343: "through the modelling"*
Thanks, it is revised based on the reviewer's comment.
- 26 *Lines 347-348: "is defined as the maximum lead-time. The definition of maximum lead-time with ..."*
Thanks, it is revised based on the reviewer's comment.

References

- [1] Apel, H., Vorogushyn, S. & Merz, B. Brief communication: Impact forecasting could substantially improve the emergency management of deadly floods: case study July 2021 floods in Germany. *Natural Hazards and Earth System Sciences* **22**, 3005–3014 (2022).
- [2] Bahra, A. Managing work flows with ecFlow. *ECMWF Newsletter* **129**, 30–32 (2011). URL <https://www.ecmwf.int/en/eLibrary/80182-managing-work-flows-ecflow>.
- [3] Najafi, H., Rakovec, O., Shrestha, P. K., Thober, S. & Samaniego, L. Post-assessment of ecmwf-mhm ensemble flood forecasting for 2021 summer flood in west germany. In *AGU Fall Meeting Abstracts*, vol. 2022, H35I–1221 (2022).
- [4] Thober, S. *et al.* Multi-model ensemble projections of european river floods and high flows at 1.5, 2, and 3 degrees global warming. *Environmental Research Letters* **13**, 014003 (2018).
- [5] Brunner, M. I. *et al.* Flood spatial coherence, triggers, and performance in hydrological simulations: large-sample evaluation of four streamflow-calibrated models. *Hydrology and Earth System Sciences* **25**, 105–119 (2021).
- [6] Silvestro, F. *et al.* Impact-based flash-flood forecasting system: Sensitivity to high resolution numerical weather prediction systems and soil moisture. *Journal of Hydrology* **572**, 388–402 (2019).
- [7] Vorogushyn, S., Apel, H., Kemter, M. & Thieken, A. H. Analyse der hochwassergefährdung im ahrtal unter berücksichtigung historischer hochwasser. *Hydrologie und Wasserbewirtschaftung* **66**, 244–254 (2022).
- [8] Speight, L. J., Cranston, M. D., White, C. J. & Kelly, L. Operational and emerging capabilities for surface water flood forecasting. *Wiley Interdisciplinary Reviews: Water* **8**, e1517 (2021).
- [9] Ivanov, V. Y. *et al.* Breaking down the computational barriers to real-time urban flood forecasting. *Geophysical Research Letters* **48**, e2021GL093585 (2021).
- [10] Pappenberger, F. *et al.* The monetary benefit of early flood warnings in europe. *Environmental Science & Policy* **51**, 278–291 (2015).
- [11] Copernicus EMS Mapping products, EMSR517. Available at: [EMSR517]: Bad Neuenahr-Ahrweiler: Grading Product, Monitoring 1, version 3, release 1, RTP Map 01 (2023). Accessed on: 4 October 2023.

[redacted]

Figure 3: Rapid Mapping of the Ahr flood by Copernicus Emergency Management Service[11]

Response to Referee #3 comments

Well done on a very good article. The authors present an advanced flood early warning system, demonstrating it's use using the 2021 European Summer floods and a case study in Germany. The article is very well written and detailed. I am not sure the title matches well with the content of the article. The content is quite technical and goes into detail about the approach. However, the title suggests a more high level overview of the utility of flood early warning systems. I also do not see where high- resolution is defined. I have provided some comments and suggestions below. My main recommendations is to change the title, and to comment how this work can be adapted to other areas, particularly those with less high quality data.

Response to Referee #3 comments

We thank the reviewer for their positive assessment of our work and as well for their insightful comments. We addressed all comments and provided additional information in point-by-point replies. Comments:

1 *Well-written*

Thanks for the comment.

2 *The Flood Early Warning Systems (FEWS): state-of-the-art, challenges, and future perspectives provides a really nice overview. It might be good to have an overview, possibly as a table, on what countries/authorities use for FEWS. This could help readers relate more to your example.*

Thanks for your feedback. A new table is added in the revised manuscript along with additional explanations (see lines 134-ff).

Major

1 *Title. In it's current guise it does not really match the topic of the article. Reading the current title, a reader would suggest you are writing about how important high resolution and real-time impact for forecasting are, rather than presenting a method/model demonstrating what is possible. Also what do you mean by high-resolution? I don't think you define it (only hyper resolution for hydrological models)*

We agree with you and revised the title to "Advancing a High-Resolution Impact-based Early Warning System for Riverine Flooding". We consider high-resolution for the flood inundation and impact modelling to operate and provide information on a grid of 1 to 10 meters [1]. This definition is provided in lines 123-124 of the revised manuscript.

2 *Your example is a good one using the River Ahr. However, the sort of data required for this model is not available in most of the world and mostly where the deadliest floods occur. Many readers, especially being published in such a high profile journal, may think this system can be employed everywhere which of course is not the case. Therefore, it might be good to flag the data requirements, or even better to test with inferior, global data to see how results change. Then there is quite a simple argument - You need data of X quality to give useable results. Or you could just discuss.*

We fully agree with the reviewer on the importance of quality data to be available for achieving useable results. This point has briefly been addressed in the first submission in lines 255-259. We have expanded this with additional discussions pertaining to data requirements in lines 439-ff in Discussion. We also add a sub-section in the methods to address the reviewer comment in lines 553-ff.

3 *Related to the above, I think the statement on line 256 'The near-real-time impact forecasting study performed here is a proof-of-concept that can be implemented in any other region around*

the world with reasonable run times' may need some caveating. What data and computing power are needed? Are gauge data essential? Does a flood threshold need to be defined? Actually you do start to go into this in the next sentence, but could potentially do with more clarification.

Thanks for your feedback. In the revised manuscript, we have provided details on computational resources and data requirements (see lines 553-ff).

"We would like to emphasize here that all the underlying datasets and modelling tools which have been used in this study are available freely. To develop a similar system in other regions, high-resolution terrain information (DEM) along with morphological datasets (e.g., soil, vegetation, etc) would be needed. Additionally, access to near-real-time meteorological forcings and river gauge station data for model calibration can be acquired from responsible agencies. To this end, growing availability of remote-sensing and satellite based information can provide additional opportunities to reliably establish the FEWS in data-scarce regions."

Minor

- 1 Line 30. You mention unprecedented rainfall events and associated flooding are more likely to occur. It would be good to have a discussion later in the article how these sort of events can be modelled in FEWSs, especially given cognitive biases.

Thanks for the comment. The challenge has been addressed in lines 462-ff of the revised manuscript.

- 2 Line 108 - exceeded might be more appropriate than crossed

Thanks for the comment. It has been revised accordingly.

- 3 I really like using the Technology readiness level in Figure 1. Maybe you could provide a reference, or some indication what the numbers mean in the caption to help readers not familiar with it.

Thanks for the feedback. A reference and a brief description have been incorporated into the figure caption to assist readers who may not be familiar with the Technology Readiness Level definitions.

- 4 Figure 2 - Do you think you could also put the 50 year return period to align with Rhineland-Palatine? Could be interesting.

Thanks for the feedback. The 50-year return period has been added to Figure 2; and other water level related information are provided in the supplementary materials.

- 5 It would be good to provide a brief explanation of what a lead-time map is. It took me a while to get my head around it. Visually it is very intuitive, but some explanation/definition would be helpful, especially to readers not so familiar with the technicalities.

We add the definition of lead-time map in the caption of Figure 3 "A maximum lead-time raster-based flood warning map is a geospatial representation that highlights the maximum available duration for flood preparedness and response."

- 6 Table 2 caption. I think you are missing the word inundation in there. i.e total length of roads and railways inundated

Thanks for the feedback. It is revised accordingly.

- 7 You could argue that future studies are needed to find out how your results/state-of-the-art FEWS can be best communicated.

Thank you for your feedback. This comment has been addressed in lines 467-469 in revised manuscript.

8 *Figure 4 is difficult to read. Could you make it 1 column and 3 rows, thus increasing the size of each panel.*

Thanks for the comment. We put the figure as Supplementary Figure S4 and have addressed the reviewer's suggestion.

References

- [1] Apel, H., Vorogushyn, S. & Merz, B. Brief communication: Impact forecasting could substantially improve the emergency management of deadly floods: case study July 2021 floods in Germany. *Natural Hazards and Earth System Sciences* **22**, 3005–3014 (2022).

REVIEWER COMMENTS

Reviewer #1 (Remarks to the Author):

I would like to thank the authors for their significant effort to address my comments. They removed several statements, which were over-claims of the innovations and significance of their work. I then ask myself the question: what are the remaining innovations that are significant enough to be worthy of publication in a prestigious journal as Nature Communications?

- The kind of modelling chain as described in the paper has already been reported in the literature as the authors now recognise in response to my review comment (Table 1 in authors' response document).
- The use of RIM2D was a key to enabling fast ensemble simulations in their modelling chain. However, RIM2D is not authors' innovation.
- The authors somewhat attempted to compare their approach with a similar ensemble hydrological approach but without a detailed hydrodynamical modelling component. The comparison was however rather superficial. In practice, it is common for agencies to use hydrological forecasts of water level at key river points to infer inundation extent using simple methods such as by matching with historical events or flood library maps prepared before events (often generated by using detailed hydrodynamic models). An innovation of the paper would have been a comparison between the proposed approach and this most common current practice.
- The new section "Enhancing Flood Forecast Communication for Informed Decision-Making and Risk Management" is a welcome addition. However, I do not see authors' innovation.

In conclusion, the authors have significantly improved the paper in the revision. However, I could not recommend its publication in Nature Communications because of a lack of significant innovations.

Reviewer #2 (Remarks to the Author):

This is my second review of the article "Advancing a high-resolution impact-based early warning system for riverine flooding" by Najafi et al.

Prior to stating my minor comments, I would like to express my appreciation to the authors for submitting a substantially improved manuscript, which I admit stands to the high standards of the Nature journals. The authors addressed beyond my expectations all my (major) concerns and provided a manuscript version, supported by complimentary numerical investigation, that adds value to the knowledge of the scientific community, service developers and flood first responders.

I would now strongly propose for accepting the manuscript after addressing my minor suggestions listed below. Once again, congratulations to the authors.

Best wishes,

Minor comments

Line 70: Data assimilation is commonly done based on hydrological information (snow depth and water equivalent, water level, discharge) and not on meteorological data only. Here I understand that you refer to NWP, and hence I propose you delete "data assimilation" to avoid confusions.

Line 173: After the end of this paragraph, I would propose that you add a short paragraph about the added contribution from this method related to pre-calculated maps. A very nice text is provided in

Response to Reviewer 2 (in page 4) 2nd comment (section: Comparison with pre-calculated inundation maps). I would propose to summarize this text into a paragraph of about 150 words highlighting the 4 arguments.

Line 191: "Post-event analysis has revealed..."

Line 194: "delayed responses, and even, no action at all". In addition, I wonder if you have examples of such cases to indicate.

Page 14 – Figure 2: How is the pink area (uncertainty) derived and why is it fixed in time? Is post-processed precipitation a single value for time to flood peak -17 to -5 hrs? Also, in the lower part of the figure, I cannot extract the values listed in the purple (HQ100) and blue (HQ50) boxes from the spaghetti plot (ensembles). Are these numbers correct?

Line 242: In the response to reviewers you mentioned 80mm. Which is the right value?

Line 259: Why "initially"? Did you re-calibrate the model after this event? I think the message here should be/is that the calibration period does not have many such extreme events to tailor the model parameters.

Lines 381-384: Mention here which is the benchmark for a smoother reading.

Lines 410 and 412: It should be HQextreme in order to be the same as in Figure 4.

Line 434: Is IB-FEWS defined somewhere? I propose to have it explicitly impact-based FEWS.

Line 444: NWP's still have uncertainties due...

Line 459: The term IBM is not used anymore in the text, so better to delete.

Line 579: "...Figure 6. Selecting..."

Response to Reviewer 2 – Page 7: Since Figure 2 goes to the Supplement, modify the caption to also present the metrics (KGE, alpha, beta, r), their ranges, and their optimal value.

Reviewer #3 (Remarks to the Author):

I would like to thank you for taking the time to respond in such detail to the reviewer comments from both myself and other reviewers. I have never seen such a comprehensive rebuttal! Well done.

The added details make the manuscript stronger. For example, the added detail about model set-up and how the simulations are run (I.e. selecting the ensemble) is really beneficial. A useful follow-up study would be asking decision makers what ensemble ranges they would like (you use minimum, median, maximum, 25% and 75%). Like you, I would assume this is a reasonable range but perhaps there are others, or maybe less is needed.

Reviewer 2's point about this being more of a technical paper and not being suited to Nature Communications is an interesting one. For the original version I would agree on hindsight, but I think with your edits this is less of an issue. Essentially you have provided a 'recipe' that requires a very particular set of ingredients that are hard to obtain but if they can be obtained and the conditions are right, then the result is useful. You now manage expectations and more eloquently describe the product. Plus, you often see a lack of detail in many Nature articles so I think having some detail is

actually welcome.

My only remaining concern is the use of the Copernicus EMS Rapid Mapping Service map as a benchmark. Although it is reasonable to assume this is a benchmark, the product is certainly not the 'truth'. People are finally starting to recognise this and from recent discussions I've had at meetings I would expect a glut of articles about this topic to emerge in the next couple of years. Probabilistic algorithms to extract the flood footprint are one such technique beginning to be used for instance which can help matters. From Line 430 I would add a sentence or two detailing which satellites are used to extract the flood footprint and note the limitations behind the data (mis-classification, timing etc, difficulty in classifying floods in built-up areas). Actually, the difficulty in classifying built-up areas could be impacting your skill scores, as under-classification in these areas may mean that the 75p ensemble is actually best rather than the maximum. So difficult to test, and is certainly outside the scope of this article, but perhaps briefly worth mentioning. Related to this, I do not fully understand what is meant by 'Percentage ratio of forecast damage to benchmark' is? I'm guessing 100% is a perfect score? If so I would add a brief description in the Table caption, or re-frame it as a deviation from damage in the benchmark (I.e. 103% would go to +3%). The latter suggestion may make your results seem worse than they are as 1% would change to -99% which would seem bad. More detail on how this metric is calculated would be appreciated too. Is it simply the hit rate, or does it penalise false alarms.

Response to Referee #1 comments

I would like to thank the authors for their significant effort to address my comments. They removed several statements, which were over-claims of the innovations and significance of their work. I then ask myself the question: what are the remaining innovations that are significant enough to be worthy of publication in a prestigious journal as Nature Communications?

We appreciate your valuable insights which have undoubtedly enhanced the quality of our manuscript. However, we respectfully disagree with the reviewer regarding the perceived significance of our innovative approach by emphasizing that the novelty of our work goes beyond isolated claims. For details, please refer to lines 113-152 and 460-462, outlining our key advancements compared to existing FEWS and the common practices used by responsible agencies.

- *The kind of modelling chain as described in the paper has already been reported in the literature as the authors now recognise in response to my review comment (Table 1 in authors' response document).*

We respectfully disagree with the reviewer's assertion. None of the studies referenced by the reviewer[1, 2, 3] has demonstrated the implementation of probabilistic impact forecasting at high-resolution, capable of real-time operation, and concurrently considering the dynamic evolution of floods using 2D hydrodynamic modeling. In addition, the reviewer has already acknowledged in their first review that ensemble flood impact forecasting presented in this study, is new.

- *The use of RIM2D was a key to enabling fast ensemble simulations in their modelling chain. However, RIM2D is not authors' innovation.*

We agree that the RIM2D usage enables fast ensemble simulations. Though, the numerical solver used in the core of RIM2D is not our innovation, the GPU implementation of RIM2D and integrating it with the hydrologic model mHM to provide probabilistic inundation and impact forecasts within an automatic operational chain is a novel innovation of our study and a contribution of the authors!

- *The authors somewhat attempted to compare their approach with a similar ensemble hydrological approach but without a detailed hydrodynamical modelling component. The comparison was however rather superficial. In practice, it is common for agencies to use hydrological forecasts of water level at key river points to infer inundation extent using simple methods such as by matching with historical events or flood library maps prepared before events (often generated by using detailed hydrodynamic models). An innovation of the paper would have been a comparison between the proposed approach and this most common current practice.*

Indeed, we agree that comparison between the proposed approach and the most common current practice is innovative. That is why we used the official flood hazard maps (e.g. HQextreme) used by local authorities in German federal states as the basis of our comparative analysis. Please refer to lines 156-166 and Figure 4 for details of this comparison. Importantly, our approach enhances this comparison by incorporating real-time flood dynamics for better-informed decision-making and disaster response.

- *The new section "Enhancing Flood Forecast Communication for Informed Decision-Making and Risk Management" is a welcome addition. However, I do not see authors' innovation.*

We believe that the inclusion of the mentioned section has enhanced the original manuscript, thanks to the reviewer's input in the first round. Please refer to our prior response for details regarding the innovation highlighted by the authors.

In conclusion, the authors have significantly improved the paper in the revision. However, I could not recommend its publication in Nature Communications because of a lack of significant innovations.

We appreciate the recognition of significant improvements during the revision. While we respect the reviewer's perspective, we firmly believe that our work contributes valuable insights to the scientific community, service developers, and first responders in a context of flood forecasting. The authors assert that the integration of probabilistic flood impact forecasting into practical early warning system workflows represents a significant innovation and it is rather disappointing to learn that the reviewer did not acknowledge these recent advances as presented in this work. Thus, we politely disagree with the reviewer's decision.

References

- [1] Gomez, M., Sharma, S., Reed, S. & Mejia, A. Skill of ensemble flood inundation forecasts at short-to medium-range timescales. *Journal of Hydrology* **568**, 207–220 (2019).
- [2] Schumann, G.-P. *et al.* A first large-scale flood inundation forecasting model. *Water Resources Research* **49**, 6248–6257 (2013).
- [3] Ivanov, V. Y. *et al.* Breaking down the computational barriers to real-time urban flood forecasting. *Geophysical Research Letters* **48**, e2021GL093585 (2021).

Response to Referee #2 comments

This is my second review of the article "Advancing a high-resolution impact-based early warning system for riverine flooding" by Najafi et al.

Prior to stating my minor comments, I would like to express my appreciation to the authors for submitting a substantially improved manuscript, which I admit to stands to the high standards of the Nature journals. The authors addressed beyond my expectations all my (major) concerns and provided a manuscript version, supported by complimentary numerical investigation, that adds value to the knowledge of the scientific community, service developers and flood first responders.

I would now strongly propose for accepting the manuscript after addressing my minor suggestions listed below. Once again, congratulations to the authors.

Best wishes,

Thank you for your positive feedback on our revised manuscript. Your encouragement is valued and we appreciate very much the time you have invested in providing a detailed review. We believe addressing your comments has significantly improved the manuscript. We have revised the manuscript according to your minor suggestions.

Minor comments

- 1 *Line 70: Data assimilation is commonly done based on hydrological information (snow depth and water equivalent, water level, discharge) and not on meteorological data only. Here I understand that you refer to NWP, and hence I propose you delete "data assimilation" to avoid confusions.*

Thanks for this remark. The sentence is now accordingly revised.

- 2 *Line 173: After the end of this paragraph, I would propose that you add a short paragraph about the added contribution from this method related to pre-calculated maps. A very nice text is provided in Response to Reviewer 2 (in page 4) 2nd comment (section: Comparison with pre-calculated inundation maps). I would propose to summarize this text into a paragraph of about 150 words highlighting the 4 arguments.*

Thanks for this suggestion. The points are summarized as recommended and the texts are revised accordingly in lines 156-166.

- 3 *Line 191: "Post-event analysis has revealed"*

Thanks, the text is revised based on the reviewer's comment in line 176.

- 4 *Line 194: "delayed responses, and even, no action at all". In addition, I wonder if you have examples of such cases to indicate.*

Thanks, an example is provided from the reference[1] in lines 179-182.

- 5 *Page 14 - Figure 2: How is the pink area (uncertainty) derived and why is it fixed in time? Is post-processed precipitation a single value for time to flood peak -17 to -5 hrs? Also, in the lower part of the figure, I cannot extract the values listed in the purple (HQ100) and blue (HQ50) boxes from the spaghetti plot (ensembles). Are these numbers correct?*

The actual precipitation totals were investigated in the official report of the event published by LfU[2]. In that report, seven precipitation products were investigated including post-processed DWD products RADOLAN-RW-DWD and RADOLAN-RL-DWD, precipitation radar data of the Institute for Technical-Scientific Hydrology (ITWH) (RADAR-ITWH-Ahr) that were processed and optimized for the Ahr catchment area[3] as well as by interpolated precipitation station data considering additional information of private weather stations (PWS) (Uni-Stuttgart-PWS) of the University of Stuttgart[4, 5]. We report the areal-average total precipitation

estimated over the entire contributing area of the Ahr River during the period 07/14 07:00 to 07/14 21:00 CET ranged between 85 to 119 mm[2]. These values are reflected in as pink area in Figure 2 from the LfU report[2]. Since these values are accumulated in time, they have been shown as a single fixed value.

In Figure 2, the exceedance probabilities shown in purple and blue are provided to summarize the information from all 320 ensemble member forecasts and for all 16 initializations. Accordingly, these values are based on 20 ensemble members issued for each initialization. Figure S1 further illustrates extracted values for each initialization. We added the reference to this Supplementary Figure S1 in the caption of Figure 2 to improve the clarity of the presented content.

6 *Line 242: In the response to reviewers you mentioned 80mm. Which is the right value?*

Thank you for your observation. Initially, we plotted 48-hour total precipitation with variations up to 100 mm. However, in alignment with the LfU report, we opted to present the total precipitation for the period 07/14 07:00 to 07/14 21:00 CET, with variations up to 80 mm. The sentence has been revised (lines 214 and 216) to accurately reflect the value reported in Figure 2.

7 *Line 259: Why "initially"? Did you re-calibrate the model after this event? I think the message here should be/is that the calibration period does not have many such extreme events to tailor the model parameters.*

Thank you for your feedback. We did not use a re-calibrated version to keep the model results similar to the real-time conditions (hindcast experiment) during the actual flood event. The sentence has been revised in lines 221-222 to incorporate your comment.

8 *Lines 381-384: Mention here which is the benchmark for a smoother reading.*

Thank you for your feedback. The sentence has been revised to incorporate your suggestion in line 309.

9 *Lines 410 and 412: It should be HQextreme in order to be the same as in Figure 4.*

Thank you for your feedback. The sentence has been revised to incorporate your comment in lines 339 and 341.

10 *Line 434: Is IB-FEWS defined somewhere? I propose to have it explicitly impact-based FEWS.*

Thank you for your feedback. The sentence has been revised to incorporate your comment.

11 *Line 444: NWP's still have uncertainties due . . .*

Thanks for the comment. The sentence is now revised in line 382.

12 *Line 459: The term IBM is not used anymore in the text, so better to delete.*

Thank you for your feedback. The sentence has been revised to incorporate your comment.

13 *Line 579: "... Figure 6. Selecting . . ." Thanks for the comment. It is now revised*

14 *Response to Reviewer 2 - Page 7: Since Figure 2 goes to the Supplement, modify the caption to also present the metrics (KGE, alpha, beta, r), their ranges, and their optimal value.*

Thanks, it is revised based on the reviewer's suggestion (See the revised caption of Figure 2).

References

- [1] M., S. *et al.* PERC floods following "Bernd". Tech. Rep., Zurich Insurance Company (2021).
- [2] LfU. Hochwasser im Juli 2021. Tech. Rep., Landesamt für Umwelt (LfU) Rheinland-Pfalz (2022). URL https://lfu.rlp.de/fileadmin/lfu/Wasserwirtschaft/Ahr-Katastrophe/Hochwasser_im_Juli2021.pdf.
- [3] Krämer, S. & Radtke, I. Radar hydrometeorological reconstruction of the precipitation extreme event of 14.07.2021 in the ahr and eifel region (2021). Unpublished project report on behalf of the State Office for the Environment Rhineland-Palatinate.
- [4] Bárdossy, A., Seidel, J. & El Hachem, A. The use of personal weather station observations to improve precipitation estimation and interpolation. *Hydrology and Earth System Sciences* **25**, 583–601 (2021).
- [5] Bárdossy, A. *et al.* Improving the estimation of areal precipitation by opportunistic precipitation measurements using the example of the ahr flood in july 2021. *Hydrology and Water Management* **66**, 208–214 (2022). Project report.

Response to Referee #3 comments

I would like to thank you for taking the time to respond in such detail to the reviewer comments from both myself and other reviewers. I have never seen such a comprehensive rebuttal! Well done.

We thank the reviewer for their positive assessment of our work and as well for their insightful comments. We addressed all comments and provided additional information in point-by-point replies.

- 1 *The added details make the manuscript stronger. For example, the added detail about model set-up and how the simulations are run (i.e. selecting the ensemble) is really beneficial. A useful follow-up study would be asking decision makers what ensemble ranges they would like (you use minimum, median, maximum, 25% and 75%). Like you, I would assume this is a reasonable range but perhaps there are others, or maybe less is needed.*

We appreciate your insightful suggestion for a follow-up study involving decision makers to understand their preferences regarding ensemble ranges. We agree wholeheartedly that incorporating input from stakeholders is a crucial next step, as their perspectives can greatly enhance the practical applicability of our approach. We add a sentence (lines 413-414) to emphasize on this point for future studies.

- 2 *Reviewer 2's point about this being more of a technical paper and not being suited to Nature Communications is an interesting one. For the original version I would agree on hindsight, but I think with your edits this is less of an issue. Essentially you have provided a "recipe" that requires a very particular set of ingredients that are hard to obtain but if they can be obtained and the conditions are right, then the result is useful. You now manage expectations and more eloquently describe the product. Plus, you often see a lack of detail in many Nature articles so I think having some detail is actually welcome.*

Thank you for carefully reviewing our manuscript and addressing the concerns raised by Reviewer 2. The analogy of our work to a "recipe", highlighting the need for specific ingredients, resonates well with our intention to provide a comprehensive methodology. We are pleased to learn that the revisions have effectively managed your expectations and improved the clarity surrounding the conditions in which our approach proves valuable. We strongly believe that studies akin to ours can significantly contribute to the ongoing enhancements of Forecast Early Warning Systems (FEWS) and bolster Research to Operation (R2O) procedures within national weather and hydrological services, as well as at leading centers in the field of hydrometeorological forecasting.

- 3 *My only remaining concern is the use of the Copernicus EMS Rapid Mapping Service map as a benchmark. Although it is reasonable to assume this is a benchmark, the product is certainly not the 'truth'. people are finally starting to recognise this and from recent discussions I've had at meetings I would expect a glut of articles about this topic to emerge in the next couple of years. probabilistic algorithms to extract the flood footprint are one such technique beginning to be used for instance which can help matters. From Line 430 I would add a sentence or two detailing which satellites are used to extract the flood footprint and note the limitations behind the data (mis-classification, timing etc, difficulty in classifying floods in built-up areas). Actually, the difficulty in classifying built-up areas could be impacting your skill scores, as under-classification in these areas may mean that the 75p ensemble is actually best rather than the maximum. So difficult to test, and is certainly outside the scope of this article, but perhaps briefly worth mentioning. Related to this, I do not fully understand what is meant by "Percentage ratio of forecast damage to benchmark" is? I'm guessing 100% is a perfect score? If so I would add a brief description in the Table caption, or re-frame it as a deviation from damage in the benchmark (i.e. 103% would go to +3%). The latter suggestion may make your results seem worse than they are as 1% would change to -99% which would seem*

bad. More detail on how this metric is calculated would be appreciated too. Is it simply the hit rate, or does it penalise false alarms.

Thanks for your comment. We address it in lines 362-367. We concur with the reviewer's observation that emergency rapid mapping is not a substitute for studies involving airborne data or ground campaigns. The official report on the event in question has also acknowledged the uncertainties linked to Copernicus Rapid Mapping. A project is actually planned to comprehensively assess and address these uncertainties within the Risk and Recovery Mapping (RRM) framework[1].

Thanks for your comment. We represent the building footprint from the forecasted chain and the benchmark calculated as a ratio between predicted building footprint and building footprint from benchmark. To address your comment, we have revised Table 2 along with its caption. Additionally, we have included an example in lines 311-313 to illustrate how the percentages in Table 2 can be interpreted.

References

- [1] LfU. Hochwasser im Juli 2021. Tech. Rep., Landesamt für Umwelt (LfU) Rheinland-Pfalz (2022). URL https://lfu.rlp.de/fileadmin/lfu/Wasserwirtschaft/Ahr-Katastrophe/Hochwasser_im_Juli2021.pdf.

REVIEWERS' COMMENTS

Reviewer #3 (Remarks to the Author):

Well done on your latest revisions. For me your responses are carefully crafted and enhance the paper further. I look forward to further work.